



# 1  A dataset of 10-year regional-scale soil moisture and soil temperature
# 2  measurements at multiple depths on the Tibetan Plateau

Pei Zhang[1,2], Donghai Zheng[2], Rogier van der Velde[1], Jun Wen[3], Yaoming Ma[2], Yijian Zeng[1], Xin
Wang[4], ZuoliangWang[4], Jiali Chen[2,5], and Zhongbo Su[1]
[1]Faculty of Geo-Information Science and Earth Observation (ITC), University of Twente, Enschede, 7514AE, the Netherlands
[2]State Key Laboratory of Tibetan Plateau Earth System, Environment and Resources, Institute of Tibetan Plateau Research,
Chinese Academy of Sciences, Beijing, 100101, China
[3]College of Atmospheric Sciences, Chengdu University of Information Technology, Chengdu, 610225, China
[4]Northwest Institute of Eco-Environment and Resources, Chinese Academy of Sciences, Lanzhou, 730000, China
[5]College of Earth and Environmental Sciences, Lanzhou University, Lanzhou, 730000, China
*Correspondence:* Donghai Zheng (zhengd@itpcas.ac.cn) and Zhongbo Su (z.su@utwente.nl)
**Abstract.** Soil moisture and soil temperature (SMST) are important state variables for quantifying exchange of heat and water
between land and atmosphere. Yet,   long-term regional-scale in-situ SMST measurements are scarce on the Tibetan Plateau
(TP), even fewer are available for multiple soil depths. "Tibet-Obs" is such a long-term regional-scale SMST observatory in
the TP established 10 years ago that includes three SMST monitoring networks, i.e., Maqu, Naqu, and Ngari (including Ali
and Shiquanhe), located in the cold humid area covered by short grasses, the cold semiarid area dominated by tundra, and the
cold arid area dominated by desert, respectively. This paper presents a long-term (~10 years) SMST profile dataset collected
from the Tibet-Obs, which includes the original in-situ measurements at a 15-min interval collected between 2008 and 2019
from all the three networks and the spatially upscaled data ($SM_{ups}$ and $ST_{ups}$) for the Maqu and Shiquanhe networks. The quality
of the upscaled data is proved to be good with errors that are generally better than the measured accuracy of adopted SMST
sensors. Long term analysis of the upscaled SMST profile data shows that the amplitudes of SMST variations decrease with
increasing soil depth, and the deeper soil layers present later onset of freezing and earlier start of thawing and thus shorter
freeze-thaw duration in both Maqu and Shiquanhe networks. In addition, there are notably differences noted between the
relationships of $SM_{ups}$ and $ST_{ups}$ under freezing conditions for the Maqu and Shiquanhe networks. No significant trend can be
found for the $SM_{ups}$ profile in the warm season (from May to October) for both networks that is consistent with the tendency
of precipitation. Similar finding is also found for the $ST_{ups}$ profile and air temperature in the Shiquanhe network during the
warm season. For the cold season (from November to April), a drying trend is noted for the $SM_{ups}$ above 20 cm in the Maqu
network, while no significant trend is found for those in the Shiquanhe network. Comparisons between the long-term upscaled
data and five reanalysis datasets indicate that none of current model-based products can reproduce the seasonal variations and
inter-annual trend changes of measured SMST profile dynamics in both networks. All the products underestimate the $ST_{ups}$ at
every depth, leading to earlier onset of freezing and later onset of thawing, which essentially demonstrates the current model
are not able to adequately simulate winter conditions on the TP. In short, the presented dataset would be valuable for evaluation



and improvement of long-term satellite- and model-based SMST products on the TP, enhancing the understanding of TP
hydrometeorological processes and their response to climate change. The dataset is available in the 4TU.ResearchData
repository at https://doi.org/10.4121/20141567.v1.
**1 Introduction**
Soil moisture and soil temperature (SMST) are important state variables for quantifying water, energy, and carbon exchange
processes in the soil-vegetation-atmosphere system (Zheng et al., 2018a; van der Velde et al., 2009). Quantifying the seasonal
dynamics and trend changes of the SMST is important to understand the response of hydrological cycle and vegetation
dynamics to climate change. Over the past decades, many efforts have been dedicated to obtain worldwide reliable SMST data
through in-situ measurements, remote sensing, and model simulations (Dorigo et al., 2011; Entekhabi et al., 2010; Rodell et
al., 2004). Thereinto, in-situ measurements are essential for the creation of ground reference for the validation of remote
sensing and model-based products (Colliander et al., 2017; Chen et al., 2017; Zeng et al., 2015), as well as improving model
parametrizations (Zheng et al., 2017, 2015a, b) and remote sensing retrieval algorithms (Zheng et al., 2019, 2018b). Since the
SMST measurements at a single site cannot well represent the value of a satellite pixel or model grid due to spatial variability,
several regional-scale monitoring networks were established to collect SMST measurements at regional-scale, some of which
are contributing to the International Soil Moisture Network (ISMN) (Dorigo et al., 2011, 2021).
Known as the third pole, exchange of water and energy between land and atmosphere on the Tibetan Plateau (TP) plays a
crucial role in regulating climate processes in the Northern Hemisphere and the evolution of the Asian monsoon (Wu et al.,
1998; Yao et al., 2012). Soil freeze-thaw (F/T) cycle is a typical process on the TP, which has a significant impact on the
energy exchange between land and atmosphere as well as water cycle (Zheng et al., 2017, 2018a). Knowledge on SMST
seasonal variations, trend changes and the F/T states on the TP can, therefore, contribute to a better understanding of the Asian
monsoon circulation and cryosphere changes. However, SMST monitoring networks are scarce on the TP compared to its vast
territory, and even fewer exist with a long time series measurements and/or with measurements at multiple soil depths. To our
knowledge, there are only two operational SMST observatories that provide long-term measurements at multiple soil depths
on the TP, i.e., Tibet-Obs (Tibetan Plateau observatory of plateau scale SMST) (Su et al., 2011; Zhang et al., 2021) and CTP-
SMTMN (Soil Moisture and Temperature Monitoring Network on the central TP) (Yang et al., 2013).
The Tibet-Obs is the first operational SMST observatory on the TP that started to provide SMST measurements in 2008, which
was designed to provide a representative coverage of distinct climate regimes and land surface conditions across the TP (Su et
al., 2011). The Tibet-Obs comprises three in-situ monitoring networks, i.e., Maqu, Naqu, and Ngari (including Ali and
Shiquanhe) (Fig. 1), which are respectively located in the cold humid area with cold dry winter and rainy summer covered by
short grasses, the cold semiarid area dominated by tundra, and the cold arid area dominated by desert. In the Tibet-Obs, SMST
sensors were installed at multiple depths, which facilitate the calibration/validation of satellite-based retrieval algorithms and
products, as well as the model-based SMST products. Table 1 summarizes the main applications of the Tibet-Obs SMST data



with focus on simultaneous usage of SM and ST measurements or usage of SM/ST measurements at multiple depths for the
product validations. A summary related to the usage of only surface SM data is included in Zhang et al. (2021). Based on Table
1 and the summary made in Zhang et al. (2021), it may be concluded that the Tibet-Obs data were mainly applied to evaluate
surface SM products, whereas a few studies simultaneously evaluated SM and ST products, and even less focused on the
investigation of profile dynamics using measurements at multiple depths. In addition, most of previous studies focused on a
certain short-term period (e.g., several years) while the Tibet-Obs holds SMST data for more than 10 years (Zhang et al., 2021),
and most of current satellite- and model-based products also provide long-term (e.g., $\geq$ 10 years) SMST data. Moreover,
previous assessments were mainly concentrated on estimating error metrics between SMST products and measurements, while
how well these SMST products can capture the long-term trend and variations of in-situ SMST dynamics is still unknown.
Therefore, development of a long-term dataset of SMST measurements at multiple depths based on the Tibet-Obs is essential
to comprehensively assess and improve the reliability of current SMST products regarding to seasonal variations and trend
changes, enhancing their applications to improve our understanding on changes of hydrological and cryosphere processes on
the TP.
In this paper, we present a long-term (~10 years) SMST profile dataset collected from the Tibet-Obs, which expands the surface
SM dataset introduced by Zhang et al. (2021) to include both SM and ST measurements collected at multiple depths. The
analysis of seasonal dynamics and trend as well as validation of model-based products are also extended to multiple depths for
an approximately 10-year period. In the Tibet-Obs, Decagon (now: METER Group) EC-TM/5TM probes and EM50 data
loggers were deployed for each site at multiple depths (e.g., 5, 10, 20, 40, 60 or 80 cm below the surface) to record SMST
profile measurements with a 15-minute interval. The presented SMST profile dataset includes in-situ measurements collected
between May 2008 and August 2019 for all three networks of the Tibet-Obs, and spatially upscaled data for the Maqu and
Shiquanhe networks.
The objective of this paper is two folds: 1) to describe the long-term in-situ SMST profile dataset including its generation and
validation, and 2) to demonstrate its uniqueness for evaluating model-based SMST profile products for a long-term period
(~10 years). The paper is organized as follows: Section 2 describes the in-situ SMST measurements collected from the Tibet-
Obs, as well as other data used in this research including meteorological data and model-based products. Section 3 presents
the spatial upscaling method, data pre-processing steps, statistical performance metrics, and Mann-Kendall trend test methods.
The preliminary analysis and applications of the SMST profile dataset are presented in Section 4. The information of data
availability is shown in Section 5. Finally, the conclusions are drawn in Section 6.



## 2 Data

### 2.1 Tibet-Obs network and in-situ SMST profile measurements

#### 2.1.1 Network design and instrumentation

The Tibet-Obs was originally established in 2008 and includes three regional-scale SMST monitoring networks (Fig. 1): the Maqu network at the eastern TP located in cold humid climate area, the Naqu network in the central TP located in cold semiarid climate area, and the Ngari network (including Ali and Shiquanhe) in the western TP located in cold arid climate area. Each network includes various numbers of in-situ SMST monitoring sites, and each monitoring site is configured with one Decagon EM50 data logger and several Decagon SMST probes (i.e., EC-TM and 5TM) to record SMST profile dynamics every 15-minute. The SMST probes were installed with the pins inserted in horizontal direction at multiple depths up to 80 cm (see Fig. 1f). The measured range of the ST sensor is from -40 to 60 °C at 0.1 °C resolution with ± 1 °C accuracy. The SM sensor measures liquid water content at a 0.0008 $m^3 m^{-3}$ resolution with ± 0.03 $m^3 m^{-3}$ accuracy. The accuracy of the SM sensor was further improved via a soil-specific calibration, leading to a root mean square difference (RMSD) of about 0.02 $m^3 m^{-3}$ (Dente et al., 2012). Nominally instruments maintenance, battery replacement, and data collection took place every year. Several initially established SMST monitoring sites were damaged by local people or animals, and there are more than 15 sites newly installed between 2014 and 2016 (see Figs. A1-A3). Therefore, there are only few monitoring sites that could provide long-term continuous SMST data records throughout the period from 2008 to 2019. Brief descriptions of SMST profile data records at each monitoring network are further provided in the following subsections, and additional information about the Tibet-Obs can be found in Zhang et al. (2021) and Su et al. (2011).

#### 2.1.2 Maqu network

The Maqu network is located in the headwaters of the Yellow River (33.60°-34.20°N, 101.70°-102.70°E) with a land cover dominated by short grasses. It covers a large river valley and its surroundings have elevations varying from 3400 to 3800 m above sea level (a.s.l). Its annual mean air temperature is about 1.2 °C and precipitation is around 600 mm per year. The Maqu network includes 26 SMST monitoring sites and covers an area of approximately 40 km by 80 km (Fig. 1b). There are 13 sites collecting SMST measurements at depths of 5, 10, 20, 40 and 80 cm, 4 sites with measurements at 5, 10, 20, and 40 cm, one site with measurements at 5, 10, and 20 cm, and 8 sites with measurements at 5 and 10 cm. The corresponding data length for every depth of each site is presented in Fig. A1 for every year from May 2008 to May 2019. Eight initially established monitoring sites were damaged before 2015, and 6 new sites were installed between 2014 and 2016. Fig. 2a shows further the number of available monitoring sites for collecting SMST measurements at different depths in the Maqu network for every month between 2008 and 2019. The number of available monitoring sites providing SMST measurements of 5 cm is up to 19 in 2009, which however, decreased as time progressed. The number of sites providing SMST measurements of 10 cm is comparable to that of 5 cm, but the SMST measurements at 20, 40, and 80 cm depths are considerably less. It can be found that the period between May 2010 and May 2011 contains the largest number of available monitoring sites. Among all the



sites, the CST05 and NST01 sites provide with 11 years of data the longest records of SMST measurements for depths of 5,
10, 20, 40, and 80 cm from 2008 to 2019 (see Fig. A1).

### 2.1.3 Ngari network

The Ngari network is located in the Ngari prefecture and includes the Shiquanhe and Ali networks. The land cover of the
network is dominated by desert system at elevations varying from 4200 to 4700 m a.s.l. Its annual mean air temperature is
about 7.0 °C and precipitation is less than 100 mm per year. The Shiquanhe network situated in vicinity of the Shiquanhe
county (32.36°-32.76°N, 79.75°-80.25°E), which includes 20 monitoring sites and covers an area of approximately 30 km by
40 km (Fig. 1d). There are 9 sites collecting the SMST measurements at depths of 5, 10, 20, 40, and 60 cm, 9 sites with
measurements at 5, 10, 20, and 40 cm, and 2 sites with measurements at 5, 10, and 20 cm. The corresponding data length for
every depth of each site is presented in Fig. A2 for every year from August 2010 to August 2019. Six initially established
monitoring sites were damaged before 2016, and 5 new sites were installed in 2016. Fig. 2b shows further the number of
available monitoring sites for collecting SMST measurements at different depths in the Shiquanhe network every month
between 2010 and 2019. The number of available monitoring sites providing SMST measurements of 5 cm is up to 14 in 2010,
which then decreased as time progressed until 2016 when new additional sites were installed, making the total up to 13 sites
in 2017. The number of sites proving SMST measurements of 10, 20, and 40 cm are comparable to that of 5 cm, which is,
however, significantly less for the SMST measurements at 60 cm. It can be also found that the period between August 2017
and August 2018 contains the largest number of available monitoring sites. Among all the sites, the SQ03 and SQ14 sites
provide with 10 years of data the longest records of SMST measurements for depths of 5, 10, 20, and 40 cm from 2010 to 2019
(see Fig. A2). The Ali network is located near the Ngari station for the Desert Environment Observation and Research of the
Chinese Academy of Science (NASDE/CAS) (33.30°-33.50°N, 79.60°-79.80°E). It consists of 4 monitoring sites (Fig. 1c) that
all collect the SMST measurements at depths of 5, 10, 20, 40, and 60 cm. The corresponding data length for every depth and
each site are presented in Fig. A2 for every year from August 2010 to August 2019 as well. Fig. 2c shows further the number
of available monitoring sites for collecting SMST measurements at different depths in the Ali network every month between
2010 and 2018. It can be found that the number of available monitoring sites providing SMST measurements for every depth
is generally less than 4 and the valid data records are not continuous, and thus the Ali network will not be used for further
analysis in this study.

### 2.1.4 Naqu network

The Naqu network is located in the Naqu River basin (31.20°-31.40°N, 91.75°-92.15°E) with a land cover dominated by
grassland (tundra). It covers a flat terrain with rolling hills at 4500 m a.s.l. on average. It exhibits the dry winter and rainy
summer receiving about 400 mm precipitation per year. The Naqu network includes 11 SMST monitoring sites (Fig. 1e) that
all collect the SMST measurements at around 5, 10, 20, 40, and 60 cm depths. The corresponding data length for every depth
of each site is presented in Fig. A3 for every year from June 2010 to August 2019. Three initially established monitoring sites



were damaged before 2016, and 4 new sites were installed in 2016. Fig. 2d shows further the number of available monitoring
sites for collecting SMST measurements at different depths in the Naqu network every month between 2010 and 2019. The
number of available monitoring sites providing SMST measurements for every depth is generally less than 4 before 2016,
which increased significantly after 2016 but with continuous valid data of less than 2 years. Therefore, the SMST data in the
Naqu network will also not be used for further analysis in this study.
**2.2 Meteorological data**
Precipitation and air temperature used in this study for the Maqu and Shiquanhe networks are obtained from the meteorological
dataset provided by the China Meteorological Administration (CMA). The dataset includes air pressure, air temperature,
evaporation, precipitation, relative humidity, sunshine duration, and wind speed, which were collected by the automatic
weather stations. The daily precipitation and air temperature collected at the Maqu (34.00°N, 102.08°E) and Shiquanhe
(32.50°N, 80.08°E) weather stations are used for comparison with the time series of SMST profile data, and the corresponding
monthly values are used for trend analysis. The daily precipitation is the cumulative value for the period between 20h of the
previous day and 20h of the current day in Beijing time, while the daily air temperature is the mean value. The monthly
precipitation is calculated by summing the daily precipitation, while the monthly mean air temperature is the average of daily
air temperature within each month.
**2.3 Model-based SMST products**
Basic information of selected model-based SMST products is given in Table 2, and brief descriptions of each product are
provided in the following subsections.
**2.3.1 ERA5**
The ERA5 is a reanalysis product obtained through the assimilation of as many observations as possible in the upper air and
near surface. The SMST data are available from 1979 till present, with a grid spacing of 0.25°*0.25° and a temporal resolution
of hourly. The SMST data of the top three model layers are used in this study, which represent the soil depths of 0-7, 7-28,
and 28-100 cm, respectively. The ERA5 product is available in the Climate Change Service (CSC) Climate Data Store (CDS)
at https://cds.climate.copernicus.eu/cdsapp#!/dataset/reanalysis-era5-single-levels?tab=form (last access: 27th June 2022).
More information about the ERA5 product can be found in Hersbach et al. (2020).
**2.3.2 GLDAS-2.1 CLSM**
The GLDAS-2.1 CLSM product (Global Land Data Assimilation System Version 2 Catchment Land Surface Model) is based
on simulations by the Catchment-F2.5 land surface model (LSM) performed with the Land Information System (LIS) Version
7. The SMST data are available from 2000 till present, with a grid resolution of 1.0°*1.0° and at a time interval of 3-hour. The
ST data for the depths of 0-10, 10-29, and 29-68 cm are selected in this study, and the surface SM (0-2 cm) and rootzone SM



(0-100 cm) data are also used. The GLDAS-2.1 CLSM product is available in the Goddard Earth Science Data and Information
Services Center (GES DISC) at https://disc.gsfc.nasa.gov/datasets/GLDAS_CLSM10_3H_2.1/summary (last access: 27th June
2022). More information about the GLDAS product can be found in Rodell et al. (2004).

**2.3.3 GLDAS-2.1 Noah**

The GLDAS-2.1 Noah product is based on the Noah LSM version 3.6 simulations performed with the LIS Version 7. The
SMST data are available from 2000 to present, with a grid resolution of 0.25°*0.25° and with a 3-hour interval. The SMST
data for the depths of 0-10, 10-40, and 40-100 cm are used in this study. The GLDAS-2.1 Noah product is available in the
GES DISC at https://disc.gsfc.nasa.gov/datasets/GLDAS_NOAH025_3H_2.1/summary (last access: 27th June 2022).

**2.3.4 GLDAS-2.1 VIC**

The GLDAS-2.1 VIC (Variable Infiltration Capacity) product is based on the VIC 4.1.2 LSM simulations performed with the
LIS Version 7. The coverage period, grid spacing and time interval of the SMST data are the same as the GLDAS-2.1 CLSM
product. The SMST data of the first and second model layers are selected in this study. The surface layer has a 30 cm depth,
whereas the depth of second layer varies with region that is about 30-130 cm for our study areas as can be found at
https://ldas.gsfc.nasa.gov/gldas/specifications (last access: 27th June 2022). The GLDAS-2.1 VIC product is available in the
GES DISC at https://disc.gsfc.nasa.gov/datasets/GLDAS_VIC10_3H_2.1/summary (last access: 27th June 2022).

**2.3.5 MERRA2**

The MERRA2 (Modern-Era Retrospective analysis for Research and Applications version 2) is the latest version of global
atmospheric reanalysis product, which uses the Goddard Earth Observing System Model (GEOS) version 5.12.4. The SMST
data are available from 1980 to present, with a grid size of 0.5°*0.625° and hourly interval. The ST data of the top three model
layers as well as SM data of surface (0-5 cm) and rootzone (0-100 cm) are selected in this study. The layer thicknesses of
model layers for the ST data also varies with region, which are 0-10, 10-30, and 30-70 cm for our study areas as can be found
at https://disc.gsfc.nasa.gov/datasets/M2C0NXLND_5.12.4/summary (last access: 27th Feb 2022). The MERRA2 product is
available in the GES DISC at https://disc.gsfc.nasa.gov/datasets/M2T1NXLND_5.12.4/summary (last access: 27th June 2022).
More information about the MERRA2 product can be found in Gelaro et al. (2017).

**3 Methods**

**3.1 Production and uncertainty analysis of upscaled SMST profile dataset**

Spatial upscaling is used to create regional-scale SMST data from in-situ measurements collected at individual location that
matched with the spatial domain of satellite-based and model-based products. Zhang et al. (2021) demonstrated the good



performance of the arithmetic averaging approach in upscaling the surface SM of the Tibet-Obs network, which is also adopted
in this study to obtain the regional-scale SMST profile data for Maqu and Shiquanhe.
The arithmetic averaging method assigns equal weights to each SMST monitoring site of the network, which can be formulated
as:
$$X_t^{ups} = \frac{1}{M} \sum_{i=1}^{M} X_{t.i}^{obs} \qquad (1)$$
where $t$ represents the time in days, $i$ represents the $i^{th}$ SMST monitoring site, $M$ represents the total number of monitoring
sites, $X_t^{ups}$ stands for the upscaled SMST, and $X_{t.i}^{obs}$ is the SMST measurements for the $i^{th}$ site.
Considering that the number of available SMST monitoring sites in the Tibet-Obs network generally changes with time (see
Fig. 2), Zhang et al. (2021) suggested to use only the sites that provide the longest continuous measurements to obtain the
long-term upscaled dataset. They also showed that the upscaled surface SM with input of all active monitoring sites regardless
of the continuity tends to produce an inconsistent trend. Therefore, we use the sites of Maqu and Shiquanhe networks that have
the longest records of SMST profile data from 2009 to 2019 to produce the long-term upscaled dataset. Specifically,
measurements collected from the CST05 and NST01 sites in the Maqu network are selected to produce the long-term regional-
scale SMST dataset for depths of 5, 20, 40, and 80 cm for the period between May 2009 and May 2019. The measurements at
the 10 cm are not used for the upscaling because the sensor at the 10 cm of CST05 site was changed one time in the mid of
May 2011 which leads to a discontinuity in the collected time series. As in Zhang et al. (2021), the measurements collected in
the year with the largest number of available monitoring sites, i.e., May 2010 and May 2011 for the Maqu network (see Fig.
2), are used to preliminarily quantify the uncertainty of upscaled SMST profile data, whereby the average of the measurements
at all the available sites are treated as ground reference for the Maqu network. Similarly, measurements collected from the
SQ03 and SQ14 sites in the Shiquanhe network are selected to produce the long-term regional-scale SMST dataset for depths
of 5, 10, 20, and 40 cm for the period between August 2010 and August 2019 since both sites only provide SMST profile
measurements up to 40 cm. The average of measurements collected at the period between August 2017 and August 2018 that
has the largest number of available sites are used to quantify the uncertainty of upscaled SMST data in the Shiquanhe network.
**3.2 Pre-processing of model-based products**
We select five widely-used model-based products (see Section 2.3) which contain both SM and ST profile simulations. To
make an objective evaluation of these products using the Tibet-Obs in-situ SMST data, some essential pre-processing steps
are undertaken regarding to three aspects: unify time interval and units of SMST simulations, determine number of model
grids that cover the in-situ network, and match the model layers to the depths of in-situ measurements.
The units of SM data from the GLDAS-2.1 CLSM, Noah, and VIC products is converted from "kg m$^{-2}$" to "m$^3$ m$^{-3}$", and the
units for the ERA5 and MERRA2 SM data is already with "m$^3$ m$^{-3}$". The units of ST data from all the model-based products
is converted from "K" to "°C". The hourly or 3-hour SMST data from all the products are averaged to daily values. We define
the period between 1$^{st}$ May and 31$^{st}$ October as the warm season, and the period between 1$^{st}$ November of the previous year





and 30[th] April of the following year as the cold season. The ERA5, GLDAS-2.1 CLSM and VIC SM data in the cold seasons
are excluded for the analysis in this study since their values represent the total soil water content including both liquid water
and ice content, while the in-situ SM data only provide measurements of liquid water content.
All the model grids falling into the scope of in-situ network are extracted from each product. Afterwards, the native grids of
each product are downscaled to 0.25°*0.25° sub-grid cells using a bilinear interpolation. Subsequently, the SMST data in all
the sub-grid cells falling into the scope of in-situ network are averaged to match the upscaled in-situ SMST data that represent
the regional-scale mean values of in-situ network (see Fig. B1).
To match the depths of in-situ SMST measurements, the model-based SMST data are resampled across the vertical soil profile
using the linear interpolation method. We assume that the SMST values of each model layer are representative for the mid-
point of this layer. For example, the SMST for the layer of 10-40 cm in the GLDAS-2.1 Noah product are representative for
the depth of 25 cm. The detailed calculation processes are presented in the Appendix B.
**3.3 Statistical indicator**
Four statistical indicators are used in this study for the evaluation of upscaled in-situ SMST data as well as the model-based
products, including Bias, root-mean-square-difference (RMSD), unbiased RMSD, and Pearson correlation coefficient (R).
They can be formulated as:
$\text{Bias} = \frac{\sum_{t=1}^{n}(X_t^{est} - X_t^{obs})}{N}$ (2)
$\text{RMSD} = \sqrt{\frac{\sum_{t=1}^{n}(X_t^{obs} - X_t^{est})^2}{N}}$ (3)
$\text{ubRMSD} = \sqrt{RMSD^2 - Bias^2}$ (4)
$R = \frac{\sum_{t=1}^{n}\left(X_t^{obs} - \overline{X^{obs}}\right)(X_t^{est} - \overline{X^{est}})}{\sqrt{\sum_{t=1}^{n}(X_t^{obs} - \overline{X^{obs}})^2}\sqrt{\sum_{t=1}^{n}(X_t^{est} - \overline{X^{est}})^2}}$ (5)
where $N$ denotes the number of data points. For the evaluation of upscaled in-situ SMST data, $X_t^{obs}$ represents the mean SMST
of the largest number of available monitoring sites in a certain year for each in-situ network (see Section 3.1), and $X_t^{est}$
represents the upscaled SMST based on the monitoring sites that provide the longest continuous measurements as input. For
the assessment of model-based products, $X_t^{obs}$ represents the upscaled SMST for each in-situ network, and $X_t^{est}$ represents the
SMST simulations derived from each product.
**3.4 Trend analysis**
The Mann Kendall trend test is used in this study to determine whether a trend is presented within the long-term SMST time
series derived either from the upscaled in-situ measurements or from the model-based products. The trend analysis is also
performed for the precipitation and air temperature data for comparison purposes. The trend analysis is respectively carried
out over the warm season, the cold season, and the full year. Therefore, the data points are monthly mean values of each year





for calculating seasonal statistics instead of annual mean value, and all missing data points are assigned an equal value smaller
than existed valid data points. If the trend test results show a significant upward or downward tendency, the Sen's slope
estimate method is adopted to quantify the magnitude of the tendency. A detailed description of the trend analysis process can
be found in Appendix C.

## 4 Results

Section 4.1 presents the upscaled SMST profile data for the Maqu and Shiquanhe networks spanning the 10-year period from
2010 to 2019 (see Section 3.1), as well as the analysis results for the SMST seasonal dynamics, trend test, detection of F/T
state and soil freezing characteristics at different depths. The uncertainty analysis results for the upscaled SMST profile data
are given in Section 4.2. Application of the upscaled data to evaluate the performance of model-based products is presented in
Section 4.3 to demonstrate its suitability for the evaluation of readily available SMST profile products.

### 4.1 Analysis of the upscaled SMST profile measurements

### 4.1.1 Maqu network

Figs. 3a and 3c show the time series of upscaled daily SM ($SM_{ups}$) and ST ($ST_{ups}$) at depths of 5, 20, 40, and 80 cm from
January 2010 to December 2018 for the Maqu network, respectively. The daily precipitation ($P$) and air temperature ($T_a$)
collected from the Maqu weather station (Fig. 1b) are also shown for comparison purposes. The time series of the $SM_{ups}$ at
different depths shows similar seasonal variations, with high values in warm summer with larger amounts of rainfall and low
values in cold winter with soil freezing and much smaller amounts of snowfall. The amplitudes of $SM_{ups}$ variations generally
decrease with increasing soil depth, with larger variations noted for soil layers above 20 cm, and smallest one at the deepest
depth of 80 cm. The soil layers below 20 cm are dryer than the upper layers in the warm season. The time series of the $ST_{ups}$
at different depths also show similar seasonality with peak values in summer and lowest values in winter that is in agreement
with the seasonal $T_a$ dynamics. The soil layers above 40 cm generally drop below 0 ℃ in winter, while the $ST_{ups}$ of 80 cm is
always greater than 0 ℃ throughout the year, indicating that the maximum freezing depth in the Maqu network is shallower
than 80 cm. The magnitude of $ST_{ups}$ variations also diminish with increasing soil depth.
Figs. 3b and 3d further show the $SM_{ups}$ and $ST_{ups}$ profile dynamics with 15-min interval for a single year between May 2010
and May 2011, which confirm that amplitudes of both $SM_{ups}$ and $ST_{ups}$ variations decrease with depth. The $SM_{ups}$ variations at
5 and 20 cm are comparable to each other and larger than those at 40 and 80 cm, which also show better response to the
precipitation in rainy season. Obvious diurnal cycles can be noted for the $ST_{ups}$ at 5 cm, which diminish with depth and are
virtually absent at 40 cm. The $ST_{ups}$ at 5 cm starts to drop below 0 ℃ around mid-November, leading to a sharp decrease of
surface $SM_{ups}$ due to freezing of the soil. The deeper layers gradually freeze as time progresses, and the freezing depths reach
its peak around mid-February. Later on, the soil starts thawing with a sharp increase of $SM_{ups}$ as the $ST_{ups}$ rises above 0 ℃,
and the entire soil profile is totally thawed around the mid-April. In general, both start date of soil freezing and end date of soil



thawing increase with increasing soil depth. To further explore the characteristics of F/T cycle in the Maqu network, Fig. 3e
shows the freezing start day (FSD), thawing end day (TED), and F/T duration of each year for the depths of 5, 20 and 40 cm
during the study period, and the 80 cm layer does not freeze (see Figs. 3c and 3d). The FSD is defined as the first day that the
daily ST drops below 0 ℃ along with sharp SM decrease in current year, and the TED is the last day of ST below 0 ℃ in next
year. The number of days between the FSD and TED is referred to as the F/T duration. There is no specific information of the
FSD and TED in 2017 for the depths of 5 and 20 cm due to missing data of in-situ ST measurements in this period, and the
same holds for the soil depth of 40 cm between 2015 and 2018 (see Figs 2a and A1). It can be observed that the inter-annual
variabilities of the FSD, TED, and F/T duration for each depth are within 30 days, and no significant trend is found. It also
confirms that the deeper layer generally shows late onset of freezing and an earlier start of thawing every year leading to
shorter F/T duration.
Figs. 4a and 4b show the Mann Kendall trend test and Sen's slope estimate for the 9-year (2010-2018) $SM_{ups}$ and $ST_{ups}$ at
depths of 5 and 20 cm for the Maqu network in the warm season, cold season, and full year. The trend analysis for the depth
of 40 cm is not presented since there is not long enough (< 7 years) continuous SMST time series due to missing data. The
trends of the $P$ and $T_a$ are also shown in Figs. 4a and 4b, respectively. As described in Section 3.4, the time series would present
a significant trend if the absolute value of statistic Z is greater than 1.96 in this study. The results show that no significant trend
is found for the $SM_{ups}$ at 5 and 20 cm in the warm season like the $P$. For the cold season, the $SM_{ups}$ at depths of 5 and 20 cm
show a drying trend despite the absence of a $P$ trend. Consequently, the $SM_{ups}$ at 5 and 20 cm in the full year also show a
drying trend with the Sen's slopes of -0.004 and -0.002, respectively, which is in agreement with the $P$ trend. The $ST_{ups}$ at
depth of 5 cm shows a decreasing trend in the warm season while no significant trend is found for the $T_a$ and $ST_{ups}$ at 20 cm.
In the cold season, there is no significant trend found for the $T_a$ and $ST_{ups}$ at 5 and 20 cm. For the full year, the $ST_{ups}$ at 5 cm
shows a decreasing trend with a Sen's slope of -0.08 while no significant trend found for the $ST_{ups}$ at 20 cm like the $T_a$.
Fig. 5 shows the soil freezing characteristics for the depths of 5, 20 and 40 cm for the Maqu network by plotting the $ST_{ups}$
against corresponding measured unfrozen SM for all subzero temperatures during the freezing and thawing periods in the cold
season. The power function fitting curves to the soil freezing characteristics and corresponding fitting parameters are given in
figure for both freezing and thawing periods. The difference between the soil freezing characteristics of freezing and thawing
periods is much smaller at the surface layer (i.e., 5 cm), which increases with increasing soil depth. At the deeper soil layers
(e.g., 20 and 40 cm), the freezing rate (i.e., the amount change of unfrozen SM with temperature) of unfrozen SM with
decreasing ST in the freezing period is larger than the thawing rate of ice content with increasing ST during the thawing period.
As such, the obtained parameter values of the power function fitting curves are identical to each other at the surface layer for
the freezing and thawing periods, which are different for the deeper soil layers. The obtained parameter values are also distinct
from each other at different soil layers, indicating the layering characteristics of frozen soil in the Maqu network.



### 4.1.2 Shiquanhe network

Figs. 6a and 6c show the time series of daily $SM_{ups}$ and $ST_{ups}$ at depths of 5, 10, 20, and 40 cm from January 2011 to December 2018 for the Shiquanhe network, respectively. The daily $P$ and $T_a$ collected from the Shiquanhe weather station (Fig. 1d) are also shown for comparison purposes. The $SM_{ups}$ time series at different depths display the similar seasonality to that found for the Maqu network. The amplitudes of $SM_{ups}$ variations generally decrease with increasing soil depth, with slightly larger variations noted for soil layers above 10 cm, and smallest one at the deepest depth of 40 cm. The layers above 10 cm are dryer than the deeper layers in the warm season expect for the rainy period. The time series of the $ST_{ups}$ at different depths also show the similar seasonality to that found for the Maqu network, whereas the amplitudes of $ST_{ups}$ variations are larger than those of the Maqu network and diminish with soil depth. The soil layers above 40 cm generally drop below 0 °C in winter, indicating that the maximum freezing depth in the Shiquanhe network is deeper than 40 cm.

Figs. 6b and 6d further show the $SM_{ups}$ and $ST_{ups}$ profile dynamics with 15-min interval for a single year between August 2017 and August 2018, which confirm that amplitudes of both $SM_{ups}$ and $ST_{ups}$ variations decrease with depth. The $SM_{ups}$ variations at 5 and 10 cm are comparable to each other and larger than those at 20 and 40 cm, which also show better response to the precipitation. Obvious diurnal cycles can be noted for the $ST_{ups}$ at 5, 10, and 20 cm, which diminish with depth and are almost absent at 40 cm. The $ST_{ups}$ at 5 and 10 cm starts to drop below 0 °C around early November, leading to a decrease of $SM_{ups}$ due to soil freezing. The deeper layers freeze as time progresses, and the freeze depths reach its maximum around early January. Later on, the soil starts thawing with an increase of $SM_{ups}$ when the $ST_{ups}$ rises above 0 °C, and the entire soil profile is totally thawed around mid-March. To further explore the characteristics of F/T cycles in Shiquanhe, Fig. 6e shows the FSD, TED, and F/T duration of each year for the depths of 5, 10, 20, and 40 cm during the study period. There is no specific information of the FSD and TED in 2011 and 2013 for the depth of 5 cm due to missing data of in-situ ST measurements in this period, and the same holds for the soil depths of 20 and 40 cm in 2018 (see Figs 2b and A2). In general, the FSD increases with increasing soil depth whereas the TED is comparable at each depth. It can be observed that the inter-annual variabilities of the FSD, TED, and F/T duration for each depth are within 20 days, and there is no significant trend found for them. It also confirms that the F/T cycles at 5 and 10 cm are almost the same with each other, and the deeper layers (i.e., 20 and 40 cm) generally show late onset of freezing, leading to shorter duration.

Figs. 7a and 7b show the trend analysis results for the 8-year (2011-2018) $SM_{ups}$ and $ST_{ups}$ at depths of 5, 20, and 40 cm for the Shiquanhe network in the warm season, cold season, and full year. The trends of the $P$ and $T_a$ are also shown in Fig. 7a and 7b, respectively. The results show that no significant trend is found for the $SM_{ups}$ at all three depths in the warm season, which is in agreement with the $P$ trend. Meanwhile, the $SM_{ups}$ at 5 and 20 cm also do not show a significant trend in the cold season like the $P$, whereas the $SM_{ups}$ at 40 cm shows a wetting trend. Consequently, the $SM_{ups}$ at 40 cm shows a wetting trend with a Sen's slope of 0.001 while no trend found for the $P$ and $SM_{ups}$ at 5 and 20 cm for the full year. The $ST_{ups}$ at all three depths do not show a significant trend in the warm season, while an increasing trend is found in the cold season, which is in agreement





with $T_a$ trend. For the full year, no trend is found for the $ST_{ups}$ at depths of 5 and 20 cm like $T_a$, while an increasing trend is
found for $ST_{ups}$ of 40 cm.
Fig. 8 shows the soil freezing characteristics for the depths of 5, 20 and 40 cm for the Shiquanhe network. The fitted power
functions to the soil freezing characteristics and the corresponding parameters are also given for the freezing and thawing
periods. It is observed that there is no notable difference between the soil freezing characteristic of freezing and thawing
periods at each depth. As such, the obtained parameter values of the power function fitting curves are identical for the freezing
and thawing periods. However, the obtained parameter values are distinct from each other at different soil layers, indicating
the layering characteristics of frozen soil in the Shiquanhe network.

## 4.2 Uncertainty analysis of the upscaled SMST profile dataset

The spatial upscaling data is inevitably subject to uncertainty as a result of the SMST spatial variabilities. Therefore, in this
section we quantify the uncertainties of the long-term upscaled SMST profile dataset for the Maqu and Shiquanhe networks
via comparisons to the mean of SM and ST measurements collected during the year with the largest number of active
monitoring sites that is considered as the "ground truth" (hereafter $SM_{tru}$ and $ST_{tru}$) as shown in Zhang et al. (2021) (see Section
3.1). The selected validation periods are from 16 May 2010 to 15 May 2011 and from 1 September 2017 to 31 August 2018
for the Maqu and Shiquanhe networks, respectively.
Fig. 9a shows the comparisons between the time series of $SM_{ups}$ and $SM_{tru}$ at soil depths of 5, 20, and 40 cm with 15-min
interval for the Maqu network from 16 May 2010 to 15 May 2011, and the comparisons between the $ST_{ups}$ and $ST_{tru}$ profile
dynamics are shown in Fig. 9b. The statistical performance metrics, i.e., bias, RMSD, ubRMSD, and R, computed between
the upscaled SMST and the ground truth are shown in the figure as well. In general, the variations of $SM_{ups}$ and $SM_{tru}$ are
consistent with each other at every depth as indicated by very high R values ($\geq 0.985$), yielding RMSD values of 0.025, 0.019,
and 0.030 $m^3 m^{-3}$ at the depths of 5, 20, and 40 cm, respectively. These RMSD values are comparable and even better than the
measurement accuracy (see Section 2.1), indicating the good performance for the $SM_{ups}$ profile data. The consistency between
the $ST_{ups}$ and $ST_{tru}$ variations is even better as indicated by higher R values ($\geq 0.995$) for each soil depth, yielding RMSD values
of 0.7, 0.2, and 0.3 °C at the depths of 5, 20, and 40 cm, respectively. These RMSD values are also better than the reported
accuracy of temperature measurements (see Section 2.1), implying the good performance for the $ST_{ups}$ profile data as well.
Table 3 presents further the FSD, TED, and F/T duration for 5, 20, and 40 cm soil depths estimated based on the upscaled
SMST profile data and ground truth, respectively. The estimated FSD, TED, and F/T duration are close to each other especially
at upper soil layers (e.g., 5 and 20 cm), and the noted differences for the FSD and TED are generally less than 3 days except
that of TED at 40 cm, leading to differences of not more than 4 days for the F/T duration.
Fig. 10a shows the comparisons between the time series of $SM_{ups}$ and $SM_{tru}$ at soil depths of 5, 20, and 40 cm with 15-min
interval for the Shiquanhe network from 1 September 2017 to the 31 August 2018, and the comparisons between the $ST_{ups}$ and
$ST_{tru}$ profile dynamics are shown in Fig. 10b. The statistical performance metrics are shown in the figures as well. Similar to
the Maqu network, the variations of $SM_{ups}$ and $SM_{tru}$ are consistent with each other for each soil depth as indicated by high R



values (> 0.92), yielding RMSD values of 0.011, 0.009, and 0.010 $m^3\,m^{-3}$ at the depths of 5, 20, and 40 cm, respectively. These
RMSD values are much better than the measured accuracy of adopted SM sensor (see Section 2.1), indicating the good
performance for the $SM_{ups}$ profile data. The consistence between the $ST_{ups}$ and $ST_{tru}$ variations is even better as indicated by
higher R value (≥ 0.97) for every soil depth. Table 3 presents further the FSD, TED, and F/T duration for 5, 20, and 40 cm soil
depths estimated based on the upscaled SMST profile data and ground truth, respectively. The estimated FSD, TED, and F/T
duration are close to each other especially at upper soil layers (e.g., 5 and 20 cm), and there is little difference for the FSD and
TED except that of TED at 40 cm, leading to differences of not more than 8 days for the F/T duration.

### 412 4.3 Application of the upscaled SMST profile dataset to validate model-based products

To demonstrate the uniqueness of the upscaled SMST profile dataset for validating existing products for a long-term period,
the performance of five model-based products is investigated in this section, including the ERA5, MERRA2, GLDAS-2.1
CLSM (hereafter CLSM), GLDAS-2.1 Noah (hereafter Noah), and GLDAS-2.1 VIC (hereafter VIC) (see Section 2.3). The
performance of these model-based products in capturing the SMST seasonal variations, long-term trend changes, and the F/T
cycle at depths of 5, 20, and 40 cm in the Maqu and Shiquanhe networks is evaluated. The cold season SM data of the ERA5,
CLSM, and VIC products are excluded for the analysis since their values represent the total soil water content while the in-
situ sensors can measure the liquid soil water content in frozen soil.

### 420 4.3.1 Maqu network

Figs. 11a-11c show the time series of daily average SM at soil depths of 5, 20, and 40 cm derived from the $SM_{ups}$ and the five
model-based products from January 2010 to December 2018 for the Maqu network. The error metrics, i.e., bias, RMSD,
ubRMSD, and R, computed between the five model-based SM data and the $SM_{ups}$ for the warm and cold season are listed in
Table 4. Among the five model-based products, the ERA5 SM product agrees best with the $SM_{ups}$ at 5 and 20 cm in the warm
season with the lowest RMSD values of 0.053 and 0.032 $m^3\,m^{-3}$ and the largest R values of 0.76 and 0.74, but it tends to
overestimate the $SM_{ups}$ at 40 cm with a bias of 0.108 $m^3\,m^{-3}$. Similarly, the VIC SM product is also able to capture the magnitude
of $SM_{ups}$ dynamics at 5 and 20 cm in the warm season with slightly larger RMSD values of 0.060 and 0.049 $m^3\,m^{-3}$, but also
overestimates the $SM_{ups}$ at 40 cm with a bias of 0.088 $m^3\,m^{-3}$. The other three products tend to considerably underestimate the
$SM_{ups}$ at 5 and 20 cm in the warm season, but they yield better estimates of the SM at 40 cm as indicated by smaller biases and
RMSD values. In the cold season, the Noah SM product generally captures well the $SM_{ups}$ variations at surface layer (i.e., 5
cm) but overestimates the $SM_{ups}$ at deeper layers (e.g., 20 and 40 cm), and overestimations are also found for the MERRA2
products at all the depth. The trend analysis results for the five model-based SM data are also presented in Fig. 4a. The results
show that no significant trend is found for any of five model-based SM products at every depth in the warm season, which is
in agreement with the trend of $SM_{ups}$. Both Noah and MERRA2 SM products are able to reproduce the drying trend noted for
the $SM_{ups}$ in the cold season and full year except for the Noah SM product of 5 cm.





Figs. 11d-11f show the time series of monthly average ST at soil depths of 5, 20, and 40 cm derived from the $ST_{ups}$ and the
five model-based products for the Maqu network. The corresponding error metrics computed by daily $ST_{ups}$ are listed in Table
4 as well. In general, the five model-based ST products have similar performance and can well capture the seasonal variations
of $ST_{ups}$ at every depth. However, they tend to underestimate the $ST_{ups}$ across the entire study period, and the magnitude of
underestimations generally increases with increasing soil depths. The trend analysis results for the five model-based ST data
are also presented in Fig. 4b. At the surface layer (i.e., 5 cm), only the VIC ST product shows a decreasing trend in the warm
season like the $ST_{ups}$, while no significant trend is found for other products. In the cold season, there is no significant trend
presented for the CLSM, Noah, and MERRA2 ST products at surface layer that is consistent with $ST_{ups}$, while the other two
products show a decreasing trend. For the full year, the Noah and VIC ST products are able to reproduce the decreasing trend
found for the $ST_{ups}$ of 5 cm, whereas no significant trend is found for other products. The trends for the deeper soil layers (i.e.,
20 and 40 cm depths) are consistent with each other for each model-based ST product, and there is no significant trend found
for the products in both warm and cold season like that $ST_{ups}$, expect the VIC ST product shows a decreasing trend.
Consequently, the ERA5, CLSM, and MERRA2 ST products do not show significant trend at deeper layers in the full year,
that is consistent with $ST_{ups}$, whereas the VIC product of two depths and Noah product of 20 cm show a decreasing trend for
the full year.
To further investigate the performance of five model-based products in capturing the characteristics of F/T cycle in the Maqu
network, Fig. 12 shows the FSD, TED, and F/T duration derived from the five model-based products and upscaled dataset for
each year during the study period. It can be observed that all the five mode-based products underestimate the FSD especially
at deeper depths. The FSD estimated based on the upscaled dataset generally increases with increasing depth, while those
estimates using the model-based products are close to each other at different depth. In contrast to the FSD, all the products
overestimate the TED at deeper depths. In other words, all the model-based products tend to produce earlier onset of freezing
and later onset of thawing, leading to longer F/T duration in comparison to the upscaled dataset. The soil freezing
characteristics for depths of 5, 20 and 40 cm obtained based on the Noah and MERRA2 products are shown in Fig. 5 as well.
It can be observed that the difference between the soil freezing characteristics of freezing and thawing periods generally
decreases with increasing soil depth for the two models that is inconsistent with the upscaled dataset. In comparison to the
upscaled dataset, both Noah and MERRA2 products tend to produce higher unfrozen SM values at the same subzero ST in the
freezing period, and overestimations are also found in the thawing period except that of Noah model at 5 cm. This can explain
why the two models overestimate the $SM_{ups}$ in the cold season especially at deeper depths as shown in Fig. 11.

### 4.3.2 Shiquanhe network

Figs. 13a-13c show the time series of daily average SM at soil depths of 5, 20, and 40 cm derived from the $SM_{ups}$ and the five
model-based products from January 2011 to December 2018 for the Shiquanhe network. The error metrics computed between
the five model-based SM data and the $SM_{ups}$ for the warm and cold season are listed in Table 5. Among the five model-based
SM products, the ERA5 product agrees best with the $SM_{ups}$ at 5 cm in the warm season with the lowest RMSD of 0.06 $m^3\,m^{-3}$





and largest R value of 0.80, while other products tend to overestimate the $SM_{ups}$ especially for the VIC product. Both the Noah
and MERRA2 products also overestimate the $SM_{ups}$ of 5 cm in the cold season. For the 20 and 40 cm deeper depths, all the
products systematically overestimate the $SM_{ups}$, among which the ERA5 product shows the lowest bias while the VIC product
presents the largest bias. The trend analysis results for the five model-based SM data are also presented in Fig. 7a. The results
show that no significant trend is found for the MERRA2 product at every depth throughout the year, that is consistent with the
$SM_{ups}$ of upper layers (i.e., 5 and 20 cm), whereas both CLSM and VIC products show a drying trend at each depth. At soil
depths of 5 cm, there is also no significant trend found for the ERA5 and Noah products like the $SM_{ups}$,    while the ERA5
product shows a drying trend at deeper layers (i.e. 20 and 40 cm) in the warm season, and Noah product also presents a drying
trend at deeper layers for the cold season and full year, both of which are inconsistent with those of $SM_{ups}$.
Figs. 13d-13f show the time series of monthly average ST at soil depths of 5, 20, and 40 cm derived from the $ST_{ups}$ and the five
model-based ST products from January 2011 to December 2018 for the Shiquanhe network. The corresponding error metrics
computed by daily $ST_{ups}$ are also listed in Table 5. Similar to the Maqu network, all the five model-based products well capture
the seasonal variations of $ST_{ups}$ at every depth, but they tend to underestimate the $ST_{ups}$ throughout the entire study period, and
the magnitude of underestimations also increases with increasing soil depth. Among all the products, the Noah and CLSM
products yields the lowest bias and RMSD in the warm and cold seasons, respectively, while the VIC product presents the
largest bias for both seasons. It should be noted that the Noah product is slight worse than the CLSM product in the cold
season. The trend analysis results for the five model-based ST data are also presented in Fig. 7b. The results show that all
products do not show significant trend at every depth in the warm season that is consistent with the $ST_{ups}$. In the cold season,
the ERA5, CLSM, and MERRA2 products show an increasing trend at every depth that is consistent with the $ST_{ups}$, while no
significant trend is found for the VIC product. An increasing trend is also noted for the Noah product of 5 and 20 cm despite
no trend is found at 40 cm. For the full year, only the ERA5 and MERRA2 products capture the trends of $ST_{ups}$ at all three
depths. At the depth of 5 and 20 cm, except the CLSM product, no significant trend is found for other products that is consistent
with the $ST_{ups}$. For the depth of 40 cm, besides the Noah and VIC products, an increasing trend is found for other products and
the $ST_{ups}$.
To further investigate the performance of five model-based products in capturing the characteristics of F/T cycle in the
Shiquanhe network, Fig. 14 shows the FSD, TED, and F/T duration derived from the five model-based products and upscaled
dataset for each year during the study period. Similar as the Maqu network, all the model-based products tend to produce
earlier onset of freezing and later onset of thawing at every depth, leading to underestimation of FSD and overestimation of
TED and thus longer F/T duration in comparison to the upscaled dataset. Among the five model-based products, the CLSM
product provides the closet estimates of TED and F/T duration compared to the upscaled dataset, while the VIC product
presents the worst performance. The soil freezing characteristics for the depths of 5, 20, and 40 cm obtained from the Noah
and MERRA2 products are shown in Fig. 8 as well. Similar to the Maqu network, both Noah and MERRA2 products tend to
produce higher unfrozen SM values at the same subzero ST in both freezing and thawing periods, leading to the overestimation



of SM in the cold season in comparison to the upscaled dataset (see Fig. 13), and the magnitude of overestimation increases
with increasing soil depth.

**5 Data availability**

A long-term (2008-2019) dataset of SMST at multiple depths on the TP is freely available from the 4TU.ResearchData
repository at https://doi.org/10.4121/20141567.v1 (Zhang et al., 2022). The original in-situ SMST data, the upscaled SMST
data, and the supplementary data are stored in .xlsx files. A user guide document is given to introduce the content of the dataset
and to provide the method to download online datasets used in this paper.

**6 Conclusions**

The Tibet-Obs is a long-term SMST observatory in the TP covering different representative climatic and land surface
conditions, which includes the Maqu, Naqu, and Ngari (including Ali and Shiquanhe) networks. The three networks are located
in the cold humid area covered by short grass, the polar area dominated by tundra, and the cold arid area dominated by desert,
respectively. Each network includes various numbers of in situ SMST monitoring sites, and each monitoring site is configured
with one Decagon (now: METER group) EM50 data logger and several Decagon SMST probes (i.e., EC-TM and 5TM) to
monitor SMST dynamics at multiple depths (e.g., 5, 10, 20, 40, and 60/80 cm underground) every 15-minute, which have
generally been in operation for over a decade. This paper presents a long-term (~10 years) SMST profile dataset collected from
the Tibet-Obs, which includes original in-situ measurements collected between 2008 and 2019 from all the three networks and
the spatially upscaled data ($SM_{ups}$ and $ST_{ups}$) for the Maqu and Shiquanhe networks. The uncertainty of the spatially upscaled
dataset are further quantified via comparison to the average of SMST measurements collected at a certain year having the
largest number of available valid monitoring sites, i.e., ground truth ($SM_{tru}$ and $ST_{tru}$). The results show that the $SM_{ups}$ and
$SM_{tru}$ are consistent with each other at every depth for both Maqu and Shiquanhe networks, yielding RMSD values that are
better than the measured accuracy of adopted SM sensor. The variations of $ST_{ups}$ also agree well with the $ST_{tru}$, and the obtained
RMSD value is also better than the measured accuracy of adopted ST sensor in the Maqu network. Therefore, it can be
concluded that the quality of the upscaled dataset is generally good.
Based on the upscaled dataset, the analysis on the seasonal variations and inter-annual trend changes of profile SMST
dynamics, as well as the characteristics of F/T cycle in an approximately 10-year period is carried out for the two
hydrometeorologically contrasting networks. The results show that the time series of both $SM_{ups}$ and $ST_{ups}$ at each depth display
notable seasonality with peak values in warm summer and lowest values in cold winter, and the amplitudes of their variations
generally decrease with increasing soil depth for both networks. It can be noted that the amplitudes of the seasonal $SM_{upas}$
variations in the cold-humid Maqu network area are larger than those of the cold-arid Shiquanhe network, whereas the $ST_{ups}$
seasonality is generally stronger within the Shinquanhe measurements. The Mann Kendall trend analysis results demonstrate



that no significant trend is found for the $SM_{ups}$ profile in the warm season (from May to October) for both networks that is
consistent with the precipitation ($P$) trend. A similar finding is also found for the $ST_{ups}$ profile and air temperature $T_a$ for the
Shiquanhe network during the warm season. For the cold season (from November to April) and the full year, a drying trend is
noted for the $SM_{ups}$ above 20 cm in the Maqu network, while no significant trend is found for those in the Shiquanhe network.
In general, the deeper soil layers in both networks present later onset of freezing and earlier thawing and thus shorter F/T
duration in comparison to the surface layer. The obtained parameter values of the power function fitting curves to the soil
freezing characteristics are distinct from each other at different soil layers in both networks, confirming the layering
characteristics of frozen soil on the TP.
To demonstrate the uniqueness of the upscaled SMST profile dataset for validating existing products for a long-term period,
the performance of five model-based products is investigated. The results show that none of the model-based products can
reproduce the seasonal variations and inter-annual trend changes of profile SMST dynamics, and the characteristics of F/T
cycle obtained based on the upscaled dataset. Among the five products, only the ERA5 product captures well the seasonal
variations and trend changes of $SM_{ups}$ dynamics at surface layer (i.e., 5 cm) during the warm season in both networks, which
also provides the lowest bias for the estimations of SM above 20 cm during the warm season. All the products underestimate
the $ST_{ups}$ at every depth in both networks, whereby the Noah and ERA5 products provide better estimations in the warm season,
and the CLSM and Noah products yield better simulations for the cold season. Consequently, all the model-based products
tend to produce earlier onset of freezing and later start of thawing at every depth, leading to underestimation of FSD and
overestimation of TED and thus longer F/T duration than observed on the ground.
Overall, the Tibet-Obs SMST observatory has greatly advanced the evaluation and improvement of satellite- and model-based
SM and ST products for their applications to the TP over the past decade (see Table 1). Development of the long-term (~10
years) SMST profile dataset collected from the Tibet-Obs is urgently needed to further strengthen relevant research and could
be of value for calibration and validation of long-term satellite- or/and model-based SMST products, improving the
representation of TP hydrometeorological processes in current land surface model and satellite-based SM retrieval algorithms,
and other applications across scientific disciplines such hydrology, meteorology and climatology.
**Author contribution**
Pei Zhang, Donghai Zheng, Rogier van der Velde and Zhongbo Su designed the framework of this work. Pei Zhang performed
the computations and data analysis, and wrote the manuscript. Donghai Zheng, Rogier van der Velde, and Zhongbo Su
supervised the progress of this work and provided critical suggestions, and revised the manuscript. Zhongbo Su, Jun Wen, and
Yaoming Ma designed the setup of Tibet-Obs, Yijian Zeng, XinWang and Zuoliang Wang involved in maintaining the Tibet-
Obs and downloading the original measurements. Pei Zhang, Zuoliang Wang, and Jiali Chen organized the data.



**Competing interests**
The authors declare that they have no conflict of interest.
**Acknowledgments**
This study was supported by the National Key Research and Development Program of China (grant no. 2021YFB3900104),
the Strategic Priority Research Program of the Chinese Academy of Sciences (grant no. XDA20100103) and the National
Natural Science Foundation of China (grant nos. 41971308 and 41871273).

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



**Table 1. Summary of the applications of Tibet-Obs SMST data and corresponding findings.**

| Literature | In-situ data | Satellite- and/or model-based products/simulations | Key findings |
|---|---|---|---|
| _Simultaneous usage of SM and ST_ | | | |
| Zheng et al. (2016) | SMST at 5, 10, 20, 40, and 80 cm depths from the Maqu network, period between 2009 and 2010. | SMST simulations by the Noah model including three sets of augmentations. | The augmentations for the turbulent and soil heat transport improved the ST profile simulations, while the augmentations for the soil water flow mitigated deficiencies of SM profile simulations by Noah model. |
| Deng et al. (2020) | SMST at 5, 10, 20, and 40 cm depths from the Maqu network, period between 2010 and 2011. | SMST simulations by two versions of the Community Land Model (CLM), i.e., versions 4.5 and 5.0. | The ST simulations from both CLM model versions coincided with the in-situ measurements, while the SM simulations showed large biases. |
| Deng et al. (2021) | SMST at 5 cm depth from the Maqu network during period of 2011 and from the Ngari network during period between 2013 and 2014. | SMST simulations by the CLM5.0 that include nine experiments evaluating soil water and heat transfer parameterizations. | (i) At the Ngari network, ST simulations in all experiments generally coincided with the observations yielding RMSE within 3℃, while SM simulations in Experiment 6 (i.e., replaced soil property data, adopted virtual temperature scheme and dry surface scheme) showed the best performance.<br>(ii) At the Maqu network, ST simulations in Experiment 5 (i.e., replaced soil property data, adopted Balland and Arp scheme and dry surface scheme) showed the best performance, while SM simulations in Experiment 1 (i.e., replaced soil property data) showed the best performance. |
| _Usage of SM at multiple depths_ | | | |
| Su et al. (2013) | SM at 5, 10, 20, 40, and 80 cm depths from the Maqu network, period between 2008 and 2009; SM around 5, 10, 20, 40, and 60 cm depths from the Naqu network, period of 2008. | SM simulations by the European Centre for Medium-Range Weather Forecasts (ECMWF) based on optimum interpolation scheme and point-wise extended Kalman filter scheme, respectively. | (i) At the Naqu network, both ECMWF's SM products showed significant overestimations in the monsoon season, indicating the ECMWF model and soil texture parameter need to be improved for the cold-semiarid area on the TP.<br>(ii) At the Maqu network, both ECMWF's SM products generally showed good and comparable performance in the humid monsoon period. |
| Bhatti et al. (2013) | SM at 5, 10, 20, 40, and 80 cm depths from the Maqu network, period of 2009. | Advanced Microwave Scanning Radiometer-Earth Observing System (AMSR-E) SM product generated by the Vrije University Amsterdam and NASA. | The in-situ SM measurements at 10 cm are more suitable to validate the AMSR-E SM product. |
| Bi et al. (2016) | SM at 5, 10, 20, 40, and 80 cm depths from the Maqu network, period between 2008 and 2010. | SM products generated by CLM, Noah, Mosaic, and VIC models implemented in Global Land Data Assimilation System V1 (GLDAS-1) and Noah model adopted in GLDAS-2. | (i) The GLDAS-2 SM product did not show better performance than the GLDAS-1 products.<br>(ii) All four models can capture well the temporal variations of in-situ SM measurements but underestimated the SM values, and the Mosaic model yielded the largest bias. |



| Ju et al. (2020) | SM at 5 and 40 cm depths from the Maqu network, period between 2011 and 2012. | SM simulations by Variable Infiltration Capacity (VIC) model with assimilation of brightness temperature ($T_B$) data from the Soil Moisture and Ocean Salinity (SMOS) mission. | Assimilation of SMOS $T_B$ data improved the performance of VIC SM product indicated by reducing the root mean square difference (RMSD) for the SM at 5 cm from 0.126 to 0.087 $m^3$ $m^{-3}$, which however, had a slight positive impact for the SM at 40 cm. |
| --- | --- | --- | --- |
| Zhuang et al. (2020) | SM at 5, 10, 20, 40, and 60/80 cm depths from the Maqu, Naqu, and Ngari networks, period between 2013 and 2016. | Surface SM (SSM) data generated by using the blend method, and then rootzone SM (RZSM) data generated by Cumulative Distribution Function (CDF) matching approach and Soil Moisture Analytical Relationship (SMAR) model based on the blended SSM data. | (i) The blended SSM product constrained by in-situ SM measurements can eliminate the influence of different LSM simulations. (ii) Both SMAR model and CDF matching approach can give reliable RZSM estimates, but the performances varied from different regions, e.g., the SMAR model provided better estimates in the semi-arid area while the CDF matching approach performed slightly better in the arid area. |
| Liu et al. (2021) | SM at 5, 10, 20, and 40 cm depths from the Maqu and Ngari networks, period between 2013 and 2015. | China Meteorological Administrational Land Data Assimilation System (CLDAS) and GLDAS SM products | The CLDAS and GLDAS SM data can capture the temporal dynamics with favorable performances, expect for the GLDAS SM data at the layer of 10-40 cm |
| Usage of ST | | | |
| Wang et al. (2016) | ST at 5 cm depth from the Maqu network, period between 2008 and 2009. | ST simulations by Noah and CLM models from GLDAS-1, and by Noah model from GLDAS-2 | GLDAS-1 CLM product overestimated the ST, while both GLDAS-1 and GLDAS-2 Noah products showed underestimations although they can replicate the daily variability of in-situ ST measurements. |
| Li et al. (2019) | ST at 5 m depth from the Maqu and Ngari networks, period between 2010 and 2011. | ST simulations by Common Land Model (CoLM) implementing three different fractional vegetation cover (FVC) schemes. | (i) At the Ngari network dominated by sparse grassland or desert, ST simulations were not sensitive to FVC scheme. (ii) At the Maqu network dominated by grass, ST simulations were improved by implementing a new FVC scheme. |
| Cao et al. (2020) | ST at 5, 10, 20, and 40 cm depths from the Maqu network, period between 2008 and 2016 | ERA5-land ST product. | ERA5-land ST data showed a negative bias in the TP, and it matched better to in-situ ST measurements in permafrost regions than in non-permafrost regions. |













**Table 2. Information for the selected model-based products.**

| Product | Spatial Resolution | Temporal Resolution | Temporal Coverage | SM Stratification (cm) | ST Stratification (cm) |
|---------|-----|-----|-----|-----|-----|
| ERA5 | | Hourly | 1979 ongoing | | 0-7, 7-28, 28-100, 100-289 |
| Noah | 0.25°× 0.25° | | | | 0-10, 10-40, 40-100 |
| CLSM | | 3 Hours | 2000 ongoing | 0-2, 0-100 | 0-10, 10-29, 29-68, 68-144 |
| VIC | 1°×1° | | | | 0-30, 30-130*, 130-150* |
| MERRA2 | 0.5°×0.625° | Hourly | 1980 ongoing | 0-5*, 0-100* | 0-10*, 10-30*, 30-70*, 70-146* |

* The depth of this layer varies with region, and the value shown here is for our study area.
**Table 3. Estimation of FSD, TED, and F/T duration at soil depths of 5, 20, and 40 cm using the upscaled SMST profile dataset and**
**ground truth in the selected single year for the Maqu and Shiquanhe networks.**

| | SMST$_{ups}$ | | | SMST$_{tru}$ | | |
|---|-----|-----|-----|-----|-----|-----|
| | 5 cm | 20 cm | 40 cm | 5 cm | 20 cm | 40 cm |
| | Maqu network | | | | | |
| FSD | 19 Nov | 10 Dec | 23 Dec | 16 Nov | 8 Dec | 26 Dec |
| TED | 24 Mar | 5 Mar | 3 Mar | 23 Mar | 7 Mar | 10 Mar |
| F/T duration | 125 | 85 | 70 | 127 | 89 | 74 |
| | Shiquanhe network | | | | | |
| FSD | 14 Nov | 17 Nov | 23 Nov | 14 Nov | 18 Nov | 23 Nov |
| TED | 18 Mar | 18 Mar | 13 Mar | 18 Mar | 18 Mar | 21 Mar |
| F/T duration | 124 | 121 | 110 | 124 | 120 | 118 |


**Table 4. Statistical indicators of model-based SMST products at soil depths of 5, 20, and 40 cm for the Maqu network in the warm**
**and cold season, respectively.**

| | | Warm season | | | | Cold season | | | |
|---|---|-----|-----|-----|-----|-----|-----|-----|-----|
| | | Bias | RMSD | ubRMSD | R | Bias | RMSD | ubRMSD | R |
| | | Soil moisture | | | | | | | |
| | ERA5 | 0.036 | 0.053 | 0.039 | 0.76 | - | - | - | - |
| | GLDAS CLSM | -0.081 | 0.098 | 0.056 | 0.35 | - | - | - | - |
| 5cm | GLDAS Noah | -0.102 | 0.116 | 0.055 | 0.42 | -0.047 | 0.088 | 0.075 | 0.52 |
| | GLDAS VIC | 0.000 | 0.060 | 0.060 | 0.38 | - | - | - | - |
| | MERRA2 | -0.092 | 0.104 | 0.049 | 0.58 | 0.009 | 0.089 | 0.088 | 0.05 |
| 20cm | ERA5 | 0.016 | 0.032 | 0.027 | 0.74 | - | - | - | - |
| | GLDAS CLSM | -0.102 | 0.108 | 0.038 | 0.32 | - | - | - | - |





| | | Bias | RMSD | ubRMSD | R | Bias | RMSD | ubRMSD | R |
|---|---|---|---|---|---|---|---|---|---|
| | GLDAS Noah | -0.122 | 0.127 | 0.037 | 0.49 | -0.031 | 0.085 | 0.079 | 0.46 |
| | GLDAS VIC | -0.013 | 0.049 | 0.047 | 0.39 | - | - | - | - |
| | MERRA2 | -0.113 | 0.118 | 0.034 | 0.50 | -0.016 | 0.089 | 0.087 | 0.13 |
| | ERA5 | 0.108 | 0.111 | 0.025 | 0.69 | - | - | - | - |
| | GLDAS CLSM | -0.018 | 0.028 | 0.022 | 0.44 | - | - | - | - |
| 40cm | GLDAS Noah | -0.040 | 0.049 | 0.028 | 0.54 | 0.042 | 0.075 | 0.062 | 0.06 |
| | GLDAS VIC | 0.088 | 0.093 | 0.029 | 0.45 | - | - | - | - |
| | MERRA2 | -0.025 | 0.034 | 0.024 | 0.50 | 0.047 | 0.074 | 0.057 | 0.34 |
| | | | | Soil temperature | | | | | |
| | ERA5 | -3.5 | 3.7 | 1.1 | 0.96 | -2.4 | 3.0 | 1.8 | 0.84 |
| | GLDAS CLSM | -3.1 | 3.4 | 1.3 | 0.94 | -2.0 | 2.8 | 2.0 | 0.91 |
| 5cm | GLDAS Noah | -3.5 | 3.9 | 1.8 | 0.89 | -2.4 | 3.6 | 2.7 | 0.89 |
| | GLDAS VIC | -4.3 | 4.4 | 1.2 | 0.95 | -2.7 | 3.1 | 1.6 | 0.87 |
| | MERRA2 | -3.5 | 3.8 | 1.4 | 0.93 | -2.6 | 3.3 | 2.0 | 0.91 |
| | ERA5 | -5.0 | 5.0 | 0.7 | 0.98 | -3.2 | 3.5 | 1.4 | 0.84 |
| | GLDAS CLSM | -4.8 | 4.9 | 1.1 | 0.95 | -3.0 | 3.4 | 1.7 | 0.87 |
| 20cm | GLDAS Noah | -5.9 | 6.3 | 2.1 | 0.84 | -2.9 | 3.3 | 1.6 | 0.88 |
| | GLDAS VIC | -5.5 | 5.6 | 1.3 | 0.92 | -3.8 | 4.1 | 1.5 | 0.85 |
| | MERRA2 | -5.1 | 5.2 | 1.0 | 0.95 | -3.6 | 4.0 | 1.8 | 0.86 |
| | ERA5 | -5.3 | 5.4 | 0.8 | 0.97 | -2.8 | 3.0 | 1.2 | 0.79 |
| | GLDAS CLSM | -5.1 | 5.2 | 0.8 | 0.97 | -2.8 | 3.2 | 1.6 | 0.77 |
| 40cm | GLDAS Noah | -6.2 | 6.5 | 1.9 | 0.85 | -2.8 | 3.1 | 1.4 | 0.82 |
| | GLDAS VIC | -5.7 | 5.8 | 1.1 | 0.93 | -3.7 | 4.0 | 1.7 | 0.74 |
| | MERRA2 | -5.9 | 6.0 | 0.9 | 0.95 | -3.3 | 3.8 | 1.8 | 0.70 |


**Table 5. Same as Table 4 but for the Shiquanhe network.**

| | | Warm season | | | | Cold season | | | |
|---|---|---|---|---|---|---|---|---|---|
| | | Bias | RMSD | ubRMSD | R | Bias | RMSD | ubRMSD | R |
| | | | | Soil moisture | | | | | |
| | ERA5 | -0.001 | 0.060 | 0.060 | 0.80 | - | - | - | - |
| | GLDAS CLSM | 0.156 | 0.158 | 0.027 | 0.53 | - | - | - | - |
| 5cm | GLDAS Noah | 0.134 | 0.142 | 0.046 | 0.64 | 0.072 | 0.075 | 0.023 | 0.12 |
| | GLDAS VIC | 0.256 | 0.259 | 0.042 | 0.38 | - | - | - | - |
| | MERRA2 | 0.070 | 0.082 | 0.042 | 0.73 | 0.060 | 0.065 | 0.024 | 0.13 |
| | ERA5 | 0.084 | 0.088 | 0.026 | 0.55 | - | - | - | - |
| | GLDAS CLSM | 0.152 | 0.153 | 0.021 | 0.56 | - | - | - | - |
| 20cm | GLDAS Noah | 0.159 | 0.161 | 0.025 | 0.66 | 0.145 | 0.146 | 0.008 | 0.28 |
| | GLDAS VIC | 0.256 | 0.259 | 0.042 | 0.31 | - | - | - | - |
| | MERRA2 | 0.087 | 0.092 | 0.028 | 0.70 | 0.086 | 0.087 | 0.016 | 0.10 |
| | ERA5 | 0.107 | 0.110 | 0.021 | 0.30 | - | - | - | - |
| | GLDAS CLSM | 0.154 | 0.155 | 0.019 | 0.39 | - | - | - | - |
| 40cm | GLDAS Noah | 0.173 | 0.174 | 0.020 | 0.49 | 0.174 | 0.175 | 0.010 | -0.19 |
| | GLDAS VIC | 0.272 | 0.274 | 0.032 | 0.29 | - | - | - | - |
| | MERRA2 | 0.117 | 0.118 | 0.015 | 0.62 | 0.123 | 0.124 | 0.009 | 0.08 |
| | | | | Soil temperature | | | | | |
| | ERA5 | -5.5 | 5.8 | 1.8 | 0.95 | -6.2 | 7.0 | 3.3 | 0.83 |
| | GLDAS CLSM | -5.9 | 6.2 | 1.6 | 0.96 | -3.0 | 3.8 | 2.2 | 0.93 |
| 5cm | GLDAS Noah | -4.7 | 5.0 | 1.6 | 0.96 | -3.8 | 4.8 | 3.0 | 0.86 |
| | GLDAS VIC | -11.8 | 12.2 | 3.1 | 0.84 | -6.6 | 7.9 | 4.4 | 0.69 |
| | MERRA2 | -8.2 | 8.4 | 1.8 | 0.95 | -5.5 | 5.8 | 1.9 | 0.95 |





| | | | | | | | | | |
|---|---|---|---|---|---|---|---|---|---|
| | ERA5 | -6.6 | 6.8 | 1.7 | 0.94 | -5.8 | 6.7 | 3.3 | 0.76 |
| | GLDAS CLSM | -7.1 | 7.2 | 1.4 | 0.96 | -3.2 | 3.8 | 2.1 | 0.92 |
| 20cm | GLDAS Noah | -5.5 | 5.6 | 1.4 | 0.96 | -2.9 | 4.1 | 2.9 | 0.83 |
| | GLDAS VIC | -12.0 | 12.2 | 2.2 | 0.89 | -7.2 | 8.1 | 3.7 | 0.71 |
| | MERRA2 | -9.2 | 9.4 | 1.6 | 0.95 | -5.6 | 5.9 | 1.6 | 0.95 |
| | ERA5 | -7.5 | 7.7 | 1.5 | 0.93 | -6.1 | 6.8 | 2.9 | 0.75 |
| | GLDAS CLSM | -8.9 | 9.0 | 1.3 | 0.96 | -3.3 | 3.8 | 1.8 | 0.92 |
| 40cm | GLDAS Noah | -6.6 | 6.7 | 1.4 | 0.95 | -2.9 | 4.0 | 2.8 | 0.77 |
| | GLDAS VIC | -12.8 | 12.9 | 1.7 | 0.92 | -7.7 | 8.2 | 3.0 | 0.72 |
| | MERRA2 | -10.8 | 11.0 | 1.6 | 0.95 | -5.9 | 6.0 | 1.4 | 0.95 |



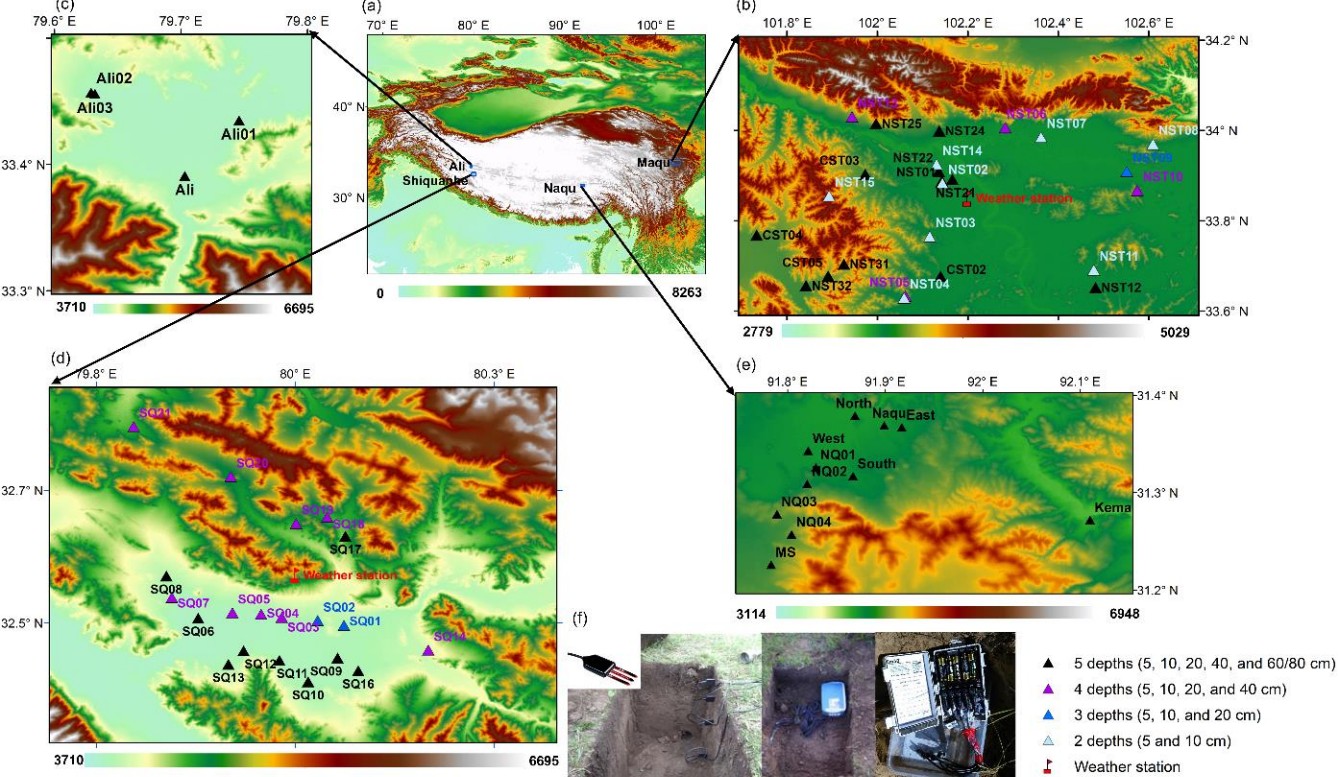


Figure 1. (a) Location of the Tibet-Obs network over the TP; Spatial distributions of SMST monitoring sites and weather station within the (b) Maqu, (c) Ali, (d) Shiquanhe, and (e) Naqu networks; and (f) an example of instruments configured for each SMST monitoring site. The triangles with different colours represent the SMST measured at different depths. (Base map is from EROS, Copyright: © EROS)




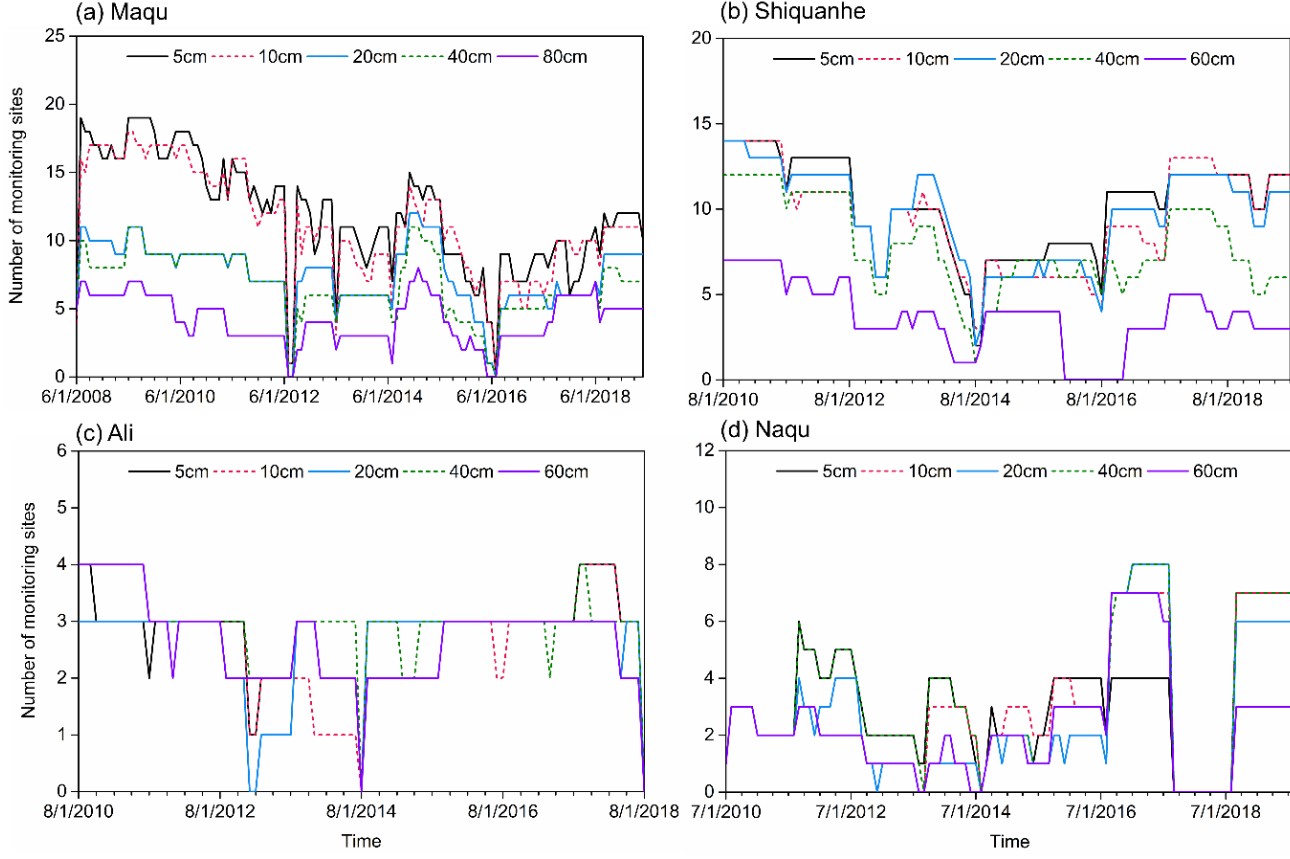


**Figure 2. Number of available SMST monitoring sites for different depths at each month for the (a) Maqu, (b) Shiquanhe, (c) Ali and (d) Naqu networks.**



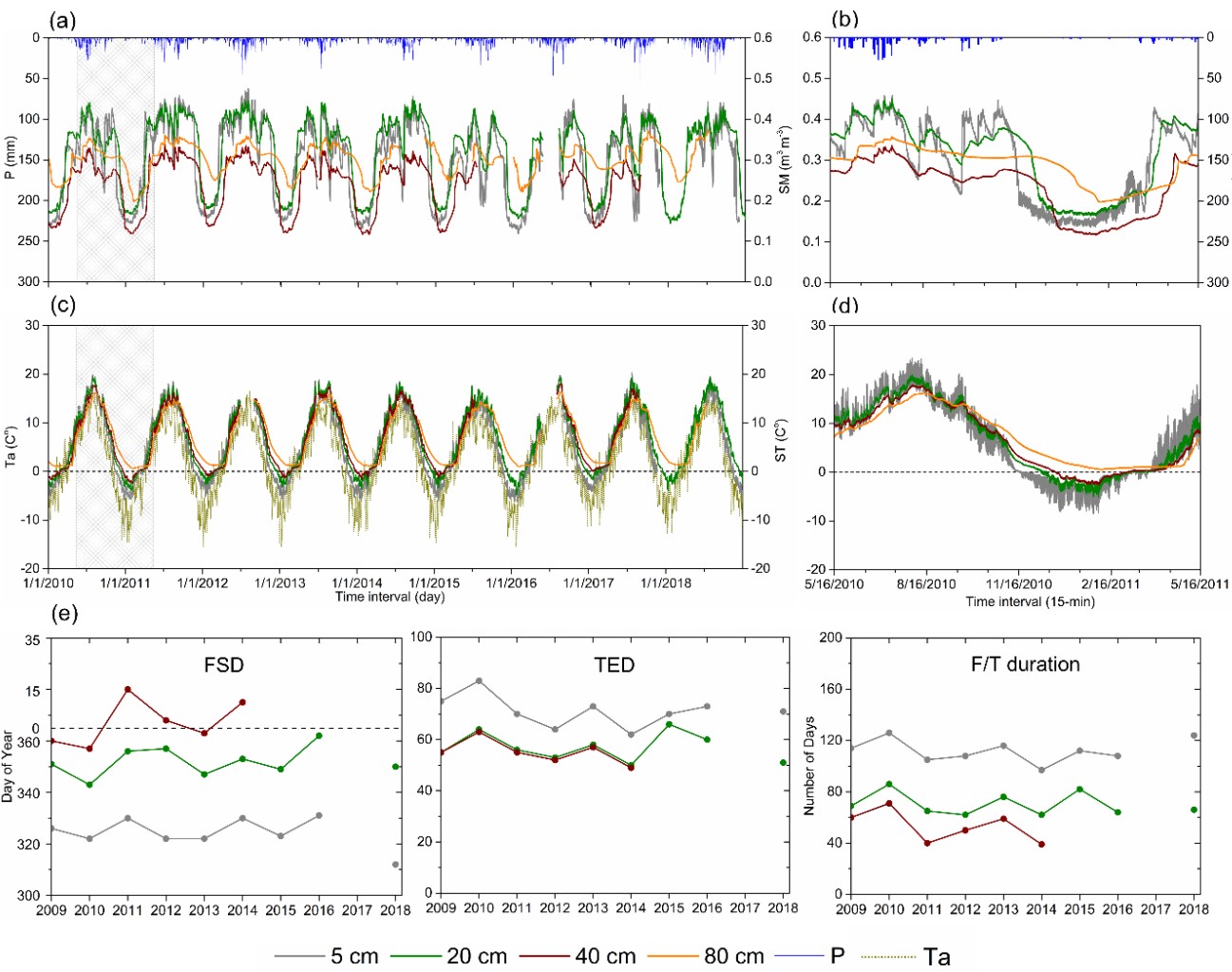


**Figure 3.** Time series of upscaled daily (a) SM$_{ups}$ and (c) ST$_{ups}$ at depths of 5, 20, 40, and 80 cm for the Maqu network between
January 2010 and December 2018; the subplots highlight the time series of upscaled (b) SM$_{ups}$ and (d) ST$_{ups}$ with interval of 15-min
between 16-5-2010 and 16-5-2011; and (e) annual variations of TSD, TED, and F/T duration at 5, 20, and 40 cm depths. The time
series of daily precipitation and air temperature are shown in (a) and (c) as well.

731

**Figure 4. Mann Kendall trend test and Sen's slope estimate for the long-term (a) SM and (b) ST at depts of 5, 20 and 40 cm from 2010 to 2018 obtained from the upscaled dataset and different model-based products for the Maqu network. The trend analysis results for the precipitation and air temperature are also shown in (a) and (b), respectively. The digits in the figure represent the values of Sen's slope estimate.**

736





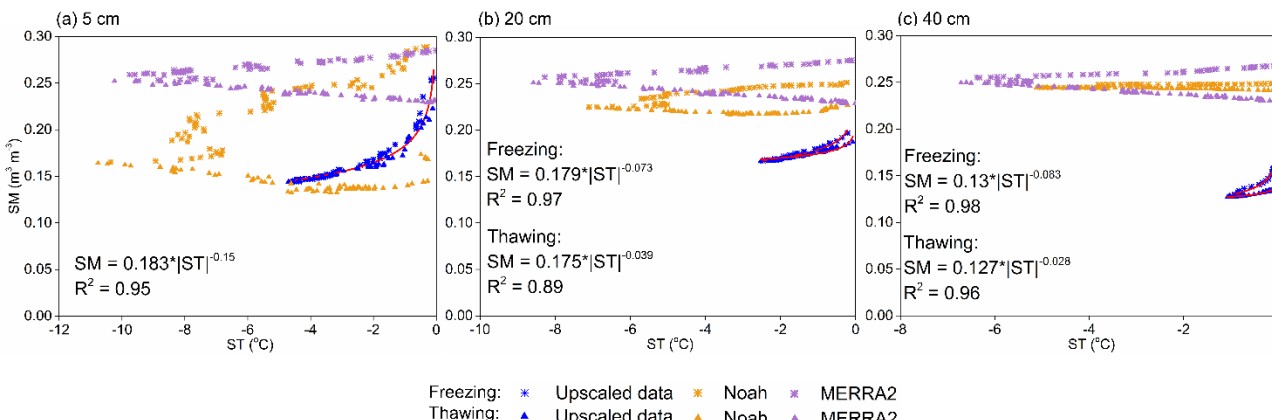

**Figure 5. Soil freezing characteristics for depths of (a) 5, (b) 20 and (c) 40 cm determined from the measured and simulated unfrozen SM and subzero ST obtained from the upscaled dataset, GLDAS Noah and MERRA2 products for the Maqu network.**

**Figure 6. Time series of upscaled daily (a) $SM_{ups}$ and (c) $ST_{ups}$ at depths of 5, 10, 20, and 40 cm for the Shiquanhe network between January 2011 and December 2018; the subplots highlight the time series of upscaled (b) $SM_{ups}$ and (d) $ST_{ups}$ with interval of 15-min between 9-1-2017 and 8-31-2018; and (e) annual variations of TSD, TED, and F/T duration at 5, 10, 20, and 40 cm depths. The time series of daily precipitation and air temperature are shown in (a) and (c) as well.**

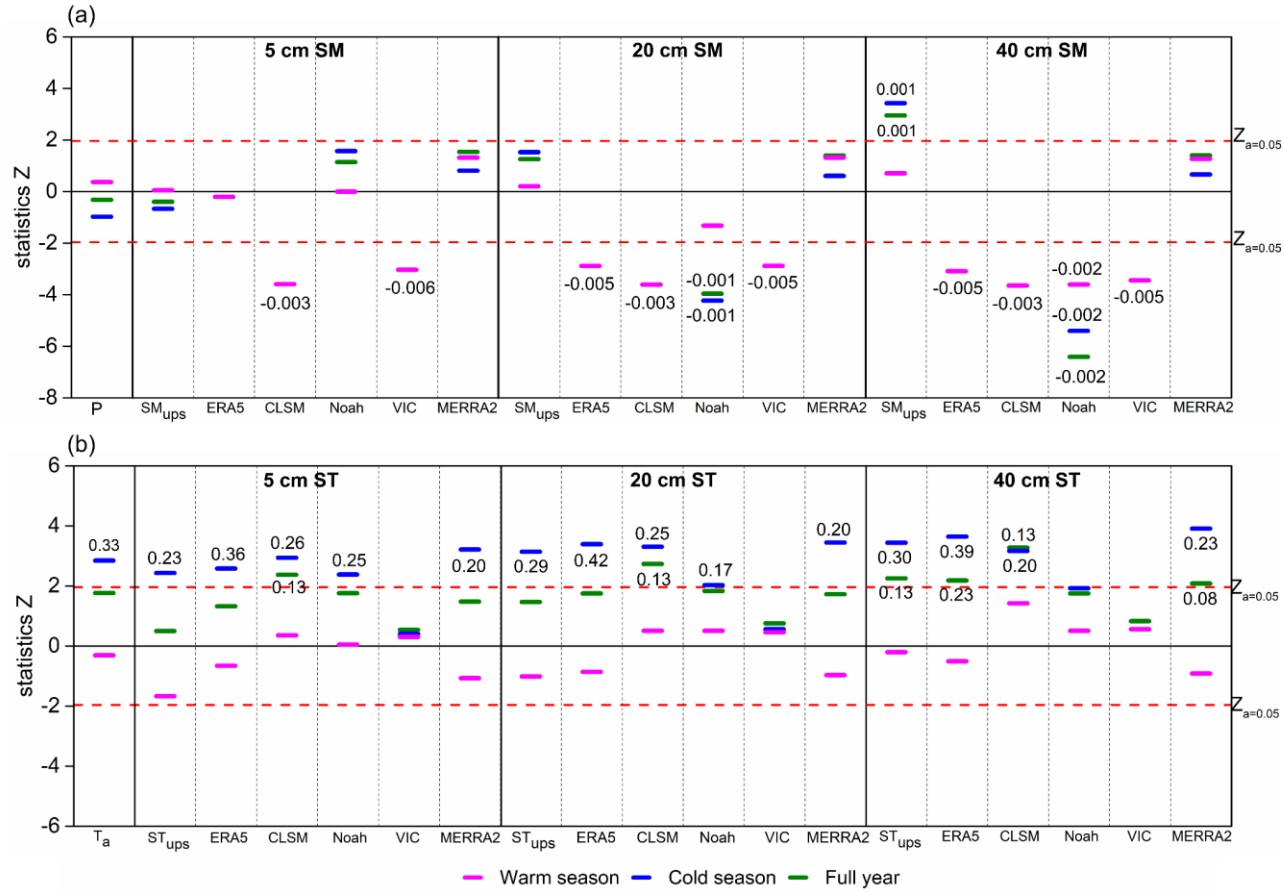

748

**Figure 7. Same as Figure 4 but for the Shiquanhe network from 2011 to 2018.**

750

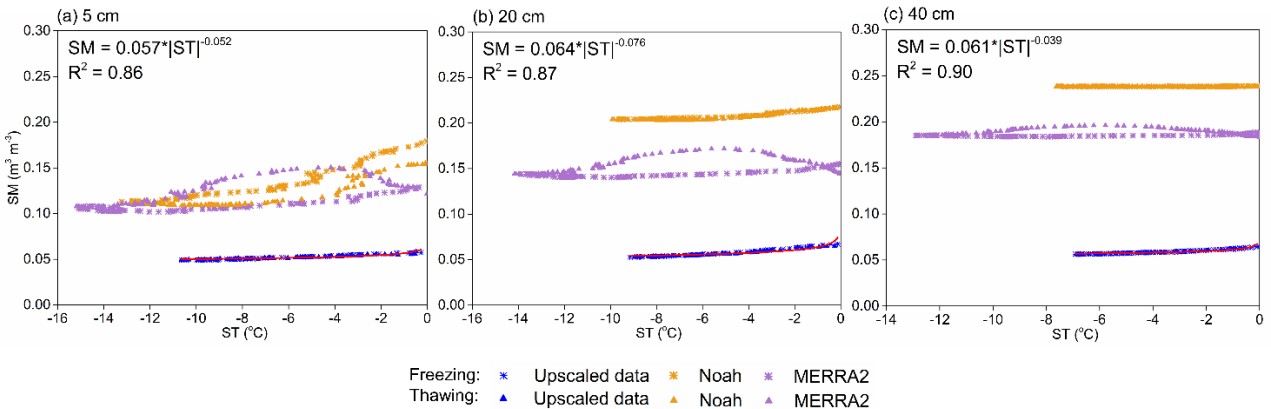

751

**Figure 8. Same as Figure 5 but for the Shiquanhe network.**



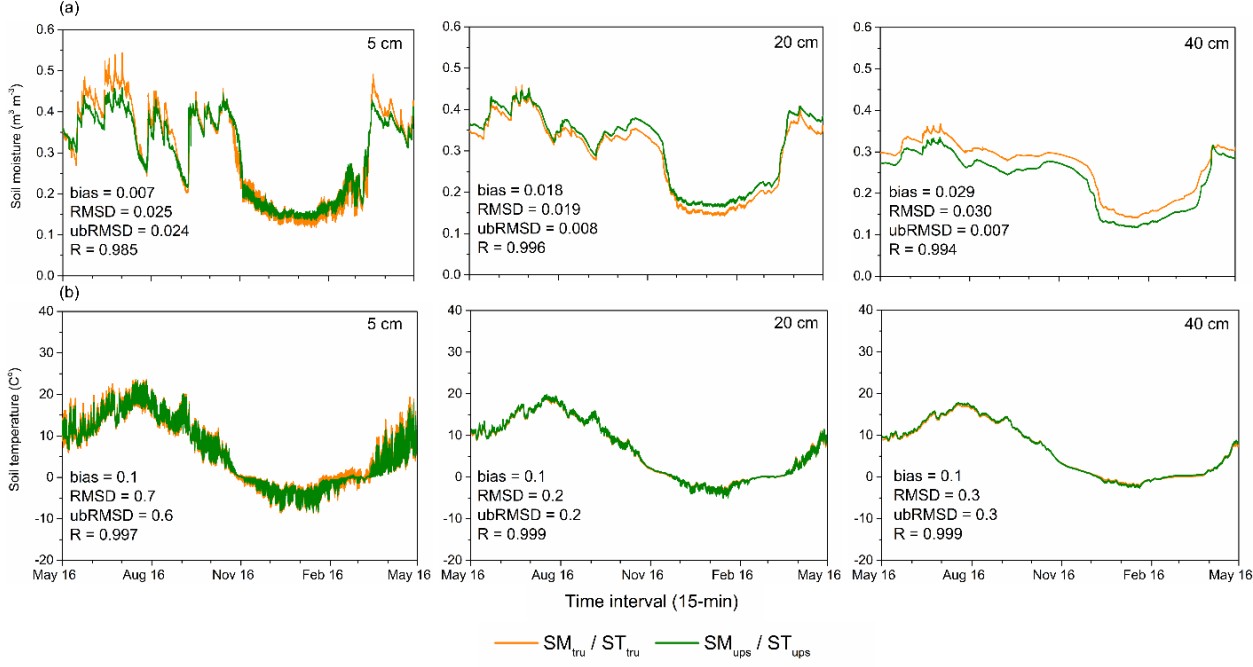

Figure 9. Comparisons between the time series of (a) SM$_{ups}$ and SM$_{tru}$, and (b) ST$_{ups}$ and ST$_{tru}$ at soil depths of 5, 20, and 40 cm with 15-min interval from 16$^{th}$ May 2010 to 16$^{th}$ May 2011 for the Maqu network.

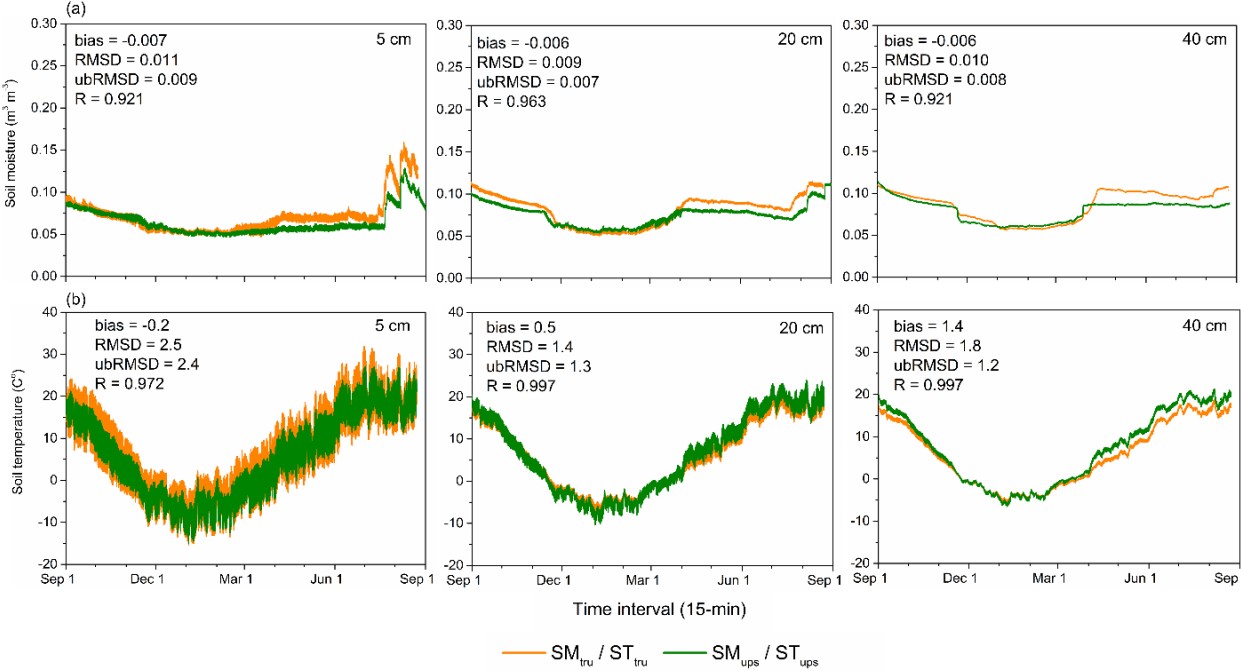

Figure 10. Same as Figure 9 but for the Shiquanhe network from 1$^{st}$ Sep 2017 and 31$^{st}$ Aug 2018.



**Figure 11. Time series of daily average SM (a-c) and monthly mean ST (d-f) at soil depths of 5 (a, d), 20 (b, e), and 40 cm (c, f) derived from the upscaled SMST dataset and five model-based products from January 2010 to December 2018 for the Maqu network.**




**Figure 12. The annual variations of FSD, TED and F/T duration at the depth of (a) 5, (b) 20, and (c) 40 cm obtained from the upscaled dataset and five model-based products for the Maqu network.**








Figure 13. Same as Figure 11 but for the Shiquanhe network from January 2011 to December 2018.







**Figure 14. Same as Figure 8 but for the Shiquanhe network.**




## Appendix A: SMST data records of the Tibet-Obs

**Figure A1. Data records of the SMST measured at different depths with temporal persistence from May 2008 to May 2019 (Y-axis) for all the monitoring sites in the Maqu network (X-axis). Cells with different colours and digits represent different number of months that contain valid SMST data in each year. Blank cells indicate that there are no measurements performed. Site names with highlight and red font represent the sites used for producing the long-term (May 2009 ~ May 2019) upscaled SMST dataset, and site names only with highlight represent the sites used for generating "ground truth" for a selected year (May 2010 ~ May 2011).**







**Figure A2.** Same as Table A1 but for the Ngari network with temporal persistence from August 2010 to August 2019. Site names with highlight and red font represent the sites used for producing the long-term (August 2010 ~ August 2019) upscaled SMST dataset, and site names only with highlight represent the sites used for generating "ground truth" for a selected year (August 2017 ~ August 2018) in the Shiquanhe network.



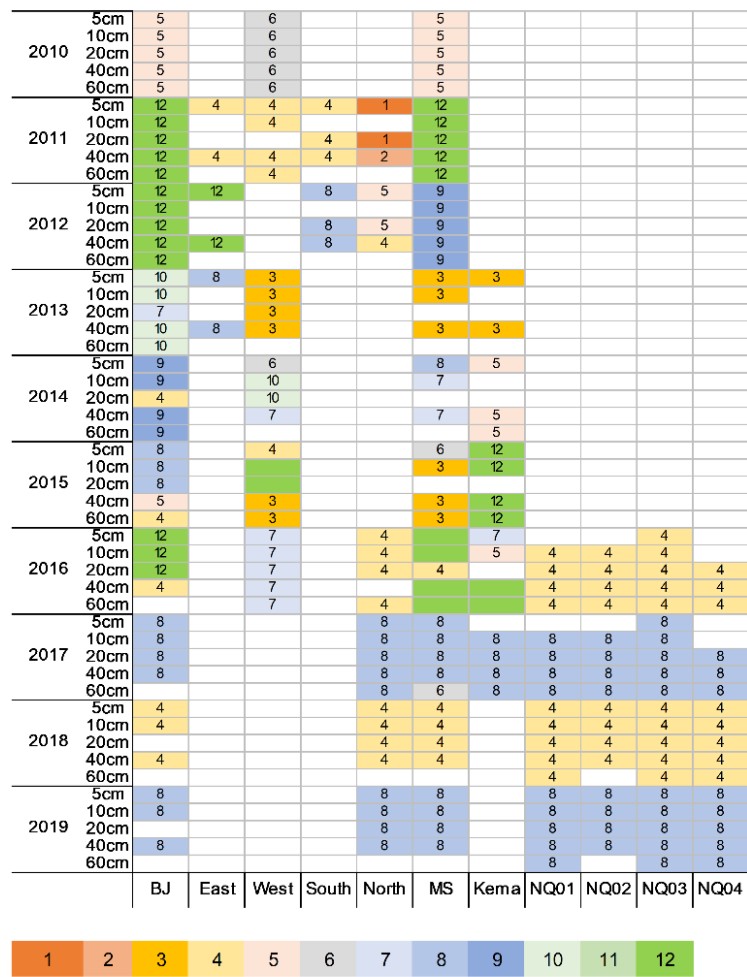


**Figure A3. Same as Table A1 but for the Naqu network with temporal persistence from June 2010 to August 2019.**







**Appendix B: Linear interpolation method for the model-based SMST data.**

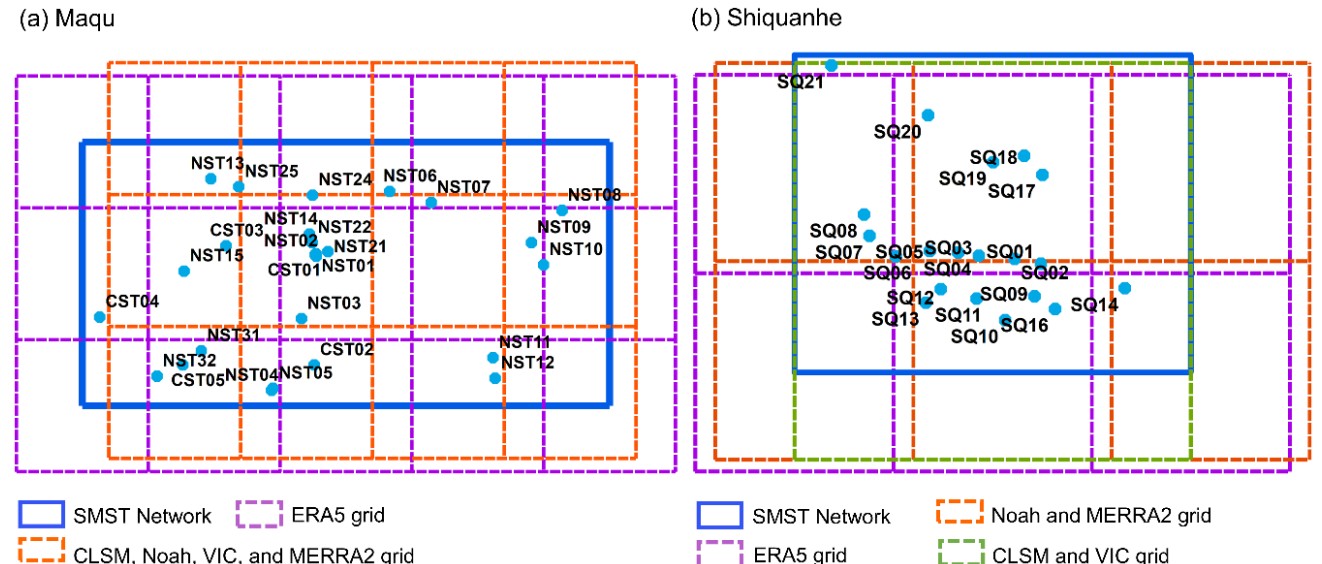

**Figure B1: Grids of the model-based products falling into the (a) Maqu and (b) Shiquanhe network areas (denoted by the colourful dashed rectangles).**

**B1 ERA5 SMST data**

The SMST derived from the ERA5 product for the depths of 5, 20, and 40 cm are calculated as:

$X_{5,ERA5} \approx X_{0-7,ERA5}$

$X_{20,ERA5} \approx X_{7-28,ERA5} + \left(X_{28-100,ERA5} - X_{7-28,ERA5}\right) * (20 - 17.5)/(64 - 17.5)$

$X_{40,ERA5} \approx X_{7-28,ERA5} + \left(X_{28-100,ERA5} - X_{7-28,ERA5}\right) * (40 - 17.5)/(64 - 17.5)$

where $X_{5,ERA5}$, $X_{20,ERA5}$, and $X_{40,ERA5}$ represent the interpolated SMST values at 5, 20, and 40 cm depths for the ERA5 product, and $X_{0-7,ERA5}$, $X_{7-28,ERA5}$, and $X_{28-100,ERA5}$ represent the SMST values for layers of 0-7, 7-28, 28-100 cm derived from the ERA5 product.

**B2 GLDAS-2.1 CLSM SMST data**

The SM derived from GLDAS-2.1 CLSM product for the depths of 5, 20, and 40 cm are calculated as:

$X_{5,GLDAS\ CLSM} \approx X_{0-2,GLDAS\ CLSM}$

$X_{20,GLDAS\ CLSM} \approx X_{0-2,GLDAS\ CLSM} + (X_{0-100,GLDAS\ CLSM} - X_{0-2,GLDAS\ CLSM}) * (20 - 1)/(50 - 1)$

$X_{40,GLDAS\ CLSM} \approx X_{0-2,GLDAS\ CLSM} + (X_{0-100,GLDAS\ CLSM} - X_{0-2,GLDAS\ CLSM}) * (40 - 1)/(50 - 1)$

The ST derived from GLDAS-2.1 CLSM product for the depths of 5, 20, and 40 cm are calculated as:

$X_{5,GLDAS\ CLSM} \approx X_{0-10,GLDAS\ CLSM}$



$X_{20,GLDAS\ CLSM} \approx X_{10-29,GLDAS\ CLSM} + (X_{29-68,GLDAS\ CLSM} - X_{10-29,GLDAS\ CLSM}) * (20 - 19.5)/(48.5 - 19.5)$
$X_{40,GLDAS\ CLSM} \approx X_{10-29,GLDAS\ CLSM} + (X_{29-68,GLDAS\ CLSM} - X_{10-29,GLDAS\ CLSM}) * (40 - 19.5)/(48.5 - 19.5)$

**B3 GLDAS-2.1 Noah SMST data**
The SMST derived from the GLDAS-2.1 Noah product for the depths of 5, 20, and 40 cm are calculated as:
$X_{5,GLDAS\ Noah} \approx X_{0-10,GLDAS\ Noah}$
$X_{20,GLDAS\ Noah} \approx X_{0-10,GLDAS\ Noah} + (X_{10-40,GLDAS\ Noah} - X_{0-10,GLDAS\ Noah}) * (20 - 5)/(25 - 5)$
$X_{40,GLDAS\ Noah} \approx X_{10-40,GLDAS\ Noah} + (X_{40-100,GLDAS\ Noah} - X_{10-40,GLDAS\ Noah}) * (40 - 25)/(70 - 25)$

**B4 GLDAS-2.1 VIC SMST data**
The SMST derived from the GLDAS-2.1 VIC product for the depths of 5, 20, and 40 cm are calculated as:
$X_{5,GLDAS\ VIC} \approx X_{0-30,GLDAS\ VIC}$
$X_{20,GLDAS\ VIC} \approx X_{0-30,GLDAS\ VIC} + (X_{30-130,GLDAS\ VIC} - X_{0-30,GLDAS\ VIC}) * (20 - 15)/(80 - 15)$
$X_{40,GLDAS\ VIC} \approx X_{0-30,GLDAS\ VIC} + (X_{30-130,GLDAS\ VIC} - X_{0-30,GLDAS\ VIC}) * (40 - 15)/(80 - 15)$

**B5 MERRA2 SMST data**
The SM derived from MERRA2 product for the depths of 5, 20, and 40 cm are calculated as:
$X_{5,MERRA2} \approx X_{0-5,MERRA2}$
$X_{20,MERRA2} \approx X_{0-5,MERRA2} + (X_{0-100,MERRA2} - X_{0-5,MERRA2}) * (20 - 2.5)/(50 - 2.5)$
$X_{40,MERRA2} \approx X_{0-5,MERRA2} + (X_{0-100,MERRA2} - X_{0-5,MERRA2}) * (40 - 2.5)/(50 - 2.5)$
The ST derived from MERRA2 product for the depths of 5, 20, and 40 cm are calculated as:
$X_{5,MERRA2} \approx X_{0-10,MERRA2}$
$X_{20,MERRA2} \approx X_{10-30,MERRA2}$
$X_{40,MERRA2} \approx X_{10-30,MERRA2} + (X_{30-70,MERRA2} - X_{10-30,MERRA2}) * (40 - 20)/(50 - 20)$

**Appendix C: Mann Kendall trend test and Sen's slope estimate**
Trend analysis for each time series is carried out as following steps:
1.Calculate month statistics ($S_i$)





For the $i^{th}$ month (1~12), $S_i$ is calculated as:
$S_i = \sum_{K=1}^{Y-1} \sum_{L=K+1}^{Y} sgn(X_{i,l} - X_{i,k})$
$sgn(X_{i,L} - X_{i,K}) = \begin{cases} 1 & X_{i,L} > X_{i,K} \\ 0 & X_{i,L} = X_{i,K} \\ -1 & X_{i,L} < X_{i,K} \end{cases}$
where $X_{i,L}$ and $X_{i,K}$ represent the monthly value of the data (e.g., SMST at different depths, precipitation, air temperature) for
the $K^{th}$ and $L^{th}$ year (satisfied $1 \leq K \leq$ Y-1, $K \leq L \leq Y$ ), Y represents the total number of years (e.g., 9 for the Maqu network
and 8 for the Shiquanhe network).
2.Calculate the variance of $S_i$ ($VAR(S_i)$)
For the $i^{th}$ month (1~12), $VAR(S_i)$ is calculated as:
$VAR(S_i) = \frac{1}{18} [Y(Y-1)(2Y+5) - \sum_{p=1}^{g_i} t_{i,p}(t_{i,p} - 1)(2t_{i,p} + 5)]$
where $g_i$ is the total number of equal-value data point group, and $t_{i,p}$ is the number of equal-value data point in the $p$th group.
3. Calculate the seasons statistic and its variance (S and VAR (S))
For the fully year, cold seasons, and warm seasons, S and VAR (S) are calculated as:
$S = \sum S_i$
$VAR (S) = \sum VAR (S_i)$
where $i$ denotes 1 ~12 for the full year, 5~10 for the warm season, and 1~4, 11, and 12 for the cold seasons.
4. Calculate the final statistic (Z)
The final statistics Z for the full year, cold seasons, and warm seasons is calculated as:
$Z = \begin{cases} \frac{S-1}{\sqrt{Var(S)}} & if\ S > 0 \\ 0 & if\ S = 0 \\ \frac{S+1}{\sqrt{Var(S)}} & if\ S < 0 \end{cases}$
If the final statistics $Z$ is positive (negative) and its absolute value is greater than $Z_{1-\alpha/2}$ (here $\alpha = 0.05$, $Z_{1-\alpha/2} = 1.96$), the
time series showed uptrend (downtrend) at the significance level of $\alpha$. Otherwise, there is no significant trend existed.
5. Sen's slope estimate
If there is a trend existed, we will further estimate the trend slope using Sen's method. For the $i^{th}$ month, individual slope $Q_i$ is
calculated as:
$Q_i = \frac{X_{i,L} - X_{i,K}}{L - K}$
where $i$ denotes 1 ~12 for the full year, 5~10 for the warm season, and 1~4, 11, and 12 for the cold seasons. The median value
of the $Q_i$ is considered as the Sen's trend slope.