# Peer review of "A dataset of 10-year regional-scale soil moisture and soil temperature measurements at multiple depths on the Tibetan Plateau"

_Earth System Science Data, 2022_

## Referee Comment (RC2)

[revised manuscript text omitted]

**Figure 6. Time series of upscaled daily (a) SM$_{ups}$ and (c) ST$_{ups}$ at depths of 5, 10, 20, and 40 cm for the Shiquanhe network between**
**January 2011 and December 2018; the subplots highlight the time series of upscaled (b) SM$_{ups}$ and (d) ST$_{ups}$ with interval of 15-min**
**between 9-1-2017 and 8-31-2018; and (e) annual variations of TSD, TED, and F/T duration at 5, 10, 20, and 40 cm depths. The time**
**series of daily precipitation and air temperature are shown in (a) and (c) as well.**

[Figure]

**Figure 7.** Same as Figure 4 but for the Shiquanhe network from 2011 to 2018.

[Figure]

**Figure 8.** Same as Figure 5 but for the Shiquanhe network.

[Figure]

[Figure]

**Figure 9. Comparisons between the time series of (a) SMups and SMtru, and (b) STups and STtru at soil depths of 5, 20, and 40 cm with 15-min interval from 16th May 2010 to 16th May 2011 for the Maqu network.**

[Figure]

**Figure 10. Same as Figure 9 but for the Shiquanhe network from 1st Sep 2017 and 31st Aug 2018.**

[Figure]

[Figure]

**Figure 11. Time series of daily average SM (a-c) and monthly mean ST (d-f) at soil depths of 5 (a, d), 20 (b, e), and 40 cm (c, f) derived**
**from the upscaled SMST dataset and five model-based products from January 2010 to December 2018 for the Maqu network.**

[Figure]

**Figure 12. The annual variations of FSD, TED and F/T duration at the depth of (a) 5, (b) 20, and (c) 40 cm obtained from the upscaled dataset and five model-based products for the Maqu network.**

[Figure]

[Figure]

**Figure 13. Same as Figure 11 but for the Shiquanhe network from January 2011 to December 2018.**

[Figure]

[Figure]

**Figure 14. Same as Figure 8 but for the Shiquanhe network.**

[Figure]

**Appendix A: SMST data records of the Tibet-Obs**

[Figure]

**Figure A1. Data records of the SMST measured at different depths with temporal persistence from May 2008 to May 2019 (Y-axis) for all the monitoring sites in the Maqu network (X-axis). Cells with different colours and digits represent different number of months that contain valid SMST data in each year. Blank cells indicate that there are no measurements performed. Site names with highlight and red font represent the sites used for producing the long-term (May 2009 ~ May 2019) upscaled SMST dataset, and site names only with highlight represent the sites used for generating "ground truth" for a selected year (May 2010 ~ May 2011).**

[Figure]

[Figure]

**Figure A2.** Same as Table A1 but for the Ngari network with temporal persistence from August 2010 to August 2019. Site names with highlight and red font represent the sites used for producing the long-term (August 2010 ~ August 2019) upscaled SMST dataset, and site names only with highlight represent the sites used for generating "ground truth" for a selected year (August 2017 ~ August 2018) in the Shiquanhe network.

[Figure]

[Figure]

**Figure A3. Same as Table A1 but for the Naqu network with temporal persistence from June 2010 to August 2019.**

[Figure]

**Appendix B: Linear interpolation method for the model-based SMST data.**

[Figure]

**Figure B1: Grids of the model-based products falling into the (a) Maqu and (b) Shiquanhe network areas (denoted by the colourful**
**dashed rectangles).**

**B1 ERA5 SMST data**

The SMST derived from the ERA5 product for the depths of 5, 20, and 40 cm are calculated as:

$X_{5,ERA5} \approx X_{0-7,ERA5}$

$X_{20,ERA5} \approx X_{7-28,ERA5} + \left(X_{28-100,ERA5} - X_{7-28,ERA5}\right) * (20 - 17.5)/(64 - 17.5)$

$X_{40,ERA5} \approx X_{7-28,ERA5} + \left(X_{28-100,ERA5} - X_{7-28,ERA5}\right) * (40 - 17.5)/(64 - 17.5)$

where $X_{5,ERA5}$, $X_{20,ERA5}$, and $X_{40,ERA5}$ represent the interpolated SMST values at 5, 20, and 40 cm depths for the ERA5 product, and $X_{0-7,ERA5}$, $X_{7-28,ERA5}$, and $X_{28-100,ERA5}$ represent the SMST values for layers of 0-7, 7-28, 28-100 cm derived from the

ERA5 product.

**B2 GLDAS-2.1 CLSM SMST data**

The SM derived from GLDAS-2.1 CLSM product for the depths of 5, 20, and 40 cm are calculated as:

$X_{5,GLDAS\ CLSM} \approx X_{0-2,GLDAS\ CLSM}$

$X_{20,GLDAS\ CLSM} \approx X_{0-2,GLDAS\ CLSM} + \left(X_{0-100,GLDAS\ CLSM} - X_{0-2,GLDAS\ CLSM}\right) * (20 - 1)/(50 - 1)$

$X_{40,GLDAS\ CLSM} \approx X_{0-2,GLDAS\ CLSM} + \left(X_{0-100,GLDAS\ CLSM} - X_{0-2,GLDAS\ CLSM}\right) * (40 - 1)/(50 - 1)$

The ST derived from GLDAS-2.1 CLSM product for the depths of 5, 20, and 40 cm are calculated as:

$X_{5,GLDAS\ CLSM} \approx X_{0-10,GLDAS\ CLSM}$

$X_{20,GLDAS\,CLSM} \approx X_{10-29,GLDAS\,CLSM} + (X_{29-68,GLDAS\,CLSM} - X_{10-29,GLDAS\,CLSM}) * (20 - 19.5)/(48.5 - 19.5)$

$X_{40,GLDAS\,CLSM} \approx X_{10-29,GLDAS\,CLSM} + (X_{29-68,GLDAS\,CLSM} - X_{10-29,GLDAS\,CLSM}) * (40 - 19.5)/(48.5 - 19.5)$

**B3 GLDAS-2.1 Noah SMST data**

The SMST derived from the GLDAS-2.1 Noah product for the depths of 5, 20, and 40 cm are calculated as:

$X_{5,GLDAS\,Noah} \approx X_{0-10,GLDAS\,Noah}$

$X_{20,GLDAS\,Noah} \approx X_{0-10,GLDAS\,Noah} + (X_{10-40,GLDAS\,Noah} - X_{0-10,GLDAS\,Noah}) * (20 - 5)/(25 - 5)$

$X_{40,GLDAS\,Noah} \approx X_{10-40,GLDAS\,Noah} + (X_{40-100,GLDAS\,Noah} - X_{10-40,GLDAS\,Noah}) * (40 - 25)/(70 - 25)$

**B4 GLDAS-2.1 VIC SMST data**

The SMST derived from the GLDAS-2.1 VIC product for the depths of 5, 20, and 40 cm are calculated as:

$X_{5,GLDAS\,VIC} \approx X_{0-30,GLDAS\,VIC}$

$X_{20,GLDAS\,VIC} \approx X_{0-30,GLDAS\,VIC} + (X_{30-130,GLDAS\,VIC} - X_{0-30,GLDAS\,VIC}) * (20 - 15)/(80 - 15)$

$X_{40,GLDAS\,VIC} \approx X_{0-30,GLDAS\,VIC} + (X_{30-130,GLDAS\,VIC} - X_{0-30,GLDAS\,VIC}) * (40 - 15)/(80 - 15)$

**B5 MERRA2 SMST data**

The SM derived from MERRA2 product for the depths of 5, 20, and 40 cm are calculated as:

$X_{5,MERRA2} \approx X_{0-5,MERRA2}$

$X_{20,MERRA2} \approx X_{0-5,MERRA2} + (X_{0-100,MERRA2} - X_{0-5,MERRA2}) * (20 - 2.5)/(50 - 2.5)$

$X_{40,MERRA2} \approx X_{0-5,MERRA2} + (X_{0-100,MERRA2} - X_{0-5,MERRA2}) * (40 - 2.5)/(50 - 2.5)$

The ST derived from MERRA2 product for the depths of 5, 20, and 40 cm are calculated as:

$X_{5,MERRA2} \approx X_{0-10,MERRA2}$

$X_{20,MERRA2} \approx X_{10-30,MERRA2}$

$X_{40,MERRA2} \approx X_{10-30,MERRA2} + (X_{30-70,MERRA2} - X_{10-30,MERRA2}) * (40 - 20)/(50 - 20)$

**Appendix C: Mann Kendall trend test and Sen's slope estimate**

Trend analysis for each time series is carried out as following steps:

1.Calculate month statistics $(S_i)$

[Figure]

For the $i^{th}$ month (1~12), $S_i$ is calculated as:

$S_i = \sum_{K=1}^{Y-1} \sum_{L=K+1}^{Y} sgn(X_{i,l} - X_{i,k})$

$sgn(X_{i,L} - X_{i,K}) = \begin{cases} 1 & X_{i,L} > X_{i,K} \\ 0 & X_{i,L} = X_{i,K} \\ -1 & X_{i,L} < X_{i,K} \end{cases}$

where $X_{i,L}$ and $X_{i,K}$ represent the monthly value of the data (e.g., SMST at different depths, precipitation, air temperature) for the $K^{th}$ and $L^{th}$ year (satisfied $1 \le K \le Y\text{-}1$, $K \le L \le Y$ ), Y represents the total number of years (e.g., 9 for the Maqu network and 8 for the Shiquanhe network).

2.Calculate the variance of $S_i$ ($VAR(S_i)$)

For the $i^{th}$ month (1~12), $VAR(S_i)$ is calculated as:

$VAR(S_i) = \frac{1}{18} [Y(Y-1)(2Y+5) - \sum_{p=1}^{g_i} t_{i,p}(t_{i,p} - 1)(2t_{i,p} + 5)]$

where $g_i$ is the total number of equal-value data point group, and $t_{i,p}$ is the number of equal-value data point in the $p$th group.

3. Calculate the seasons statistic and its variance (S and VAR (S))

For the fully year, cold seasons, and warm seasons, S and VAR (S) are calculated as:

$S = \sum S_i$

VAR (S) $= \sum VAR (S_i)$

where $i$ denotes 1 ~12 for the full year, 5~10 for the warm season, and 1~4, 11, and 12 for the cold seasons.

4. Calculate the final statistic (Z)

The final statistics Z for the full year, cold seasons, and warm seasons is calculated as:

$Z = \begin{cases} \frac{S-1}{\sqrt{Var(S)}} & if\ S > 0 \\ 0 & if\ S = 0 \\ \frac{S+1}{\sqrt{Var(S)}} & if\ S < 0 \end{cases}$

If the final statistics $Z$ is positive (negative) and its absolute value is greater than $Z_{1-\alpha/2}$ (here $\alpha = 0.05$, $Z_{1-\alpha/2} = 1.96$), the time series showed uptrend (downtrend) at the significance level of $\alpha$. Otherwise, there is no significant trend existed.

5. Sen's slope estimate

If there is a trend existed, we will further estimate the trend slope using Sen's method. For the $i^{th}$ month, individual slope $Q_i$ is calculated as:

$Q_i = \frac{X_{i,L} - X_{i,K}}{L - K}$

where $i$ denotes 1 ~12 for the full year, 5~10 for the warm season, and 1~4, 11, and 12 for the cold seasons. The median value of the $Q_i$ is considered as the Sen's trend slope.

---

## Author Comment (AC1)

**Letter to Editor**

Dear Editor,

Thank you very much for handling our manuscript and providing constructive remarks. This has helped us to improve our manuscript significantly. The manuscript has now been thoroughly revised and strengthened based on all the comments and recommendations made by both reviewers. Please find below our detailed responses to each comment made by the reviewers.

We think that the revised manuscript has appropriately addressed all the reviewers' concerns and we hope that you can consider it for publication in Earth System Science Data.

Sincerely,
Pei Zhang
On behalf of all co-authors

We would like to thank the reviewer for carefully reading our manuscript and providing detailed and constructive comments. This has helped us to improve our manuscript significantly. In the text below we provide our response to each comment point by point.

Reviewer's comments are in **bold**.

Author's responses are in regular.

Author's additions/modifications in the text are in blue.

**This manuscript introduces a valuable dataset (i.e., soil moisture and soil temperature at different depth from 5 cm to 80 cm) collected for nearly 10 years (from 2008 to 2019) at three dense networks including Maqu, Naqu, and Ngari built on the Tibetan Plateau. This dataset is a good extension of the surface soil moisture dataset introduced by Zhang et al. (2021, ESSD). The characteristics and trend of the in situ datasets as well as the derived freeze/thaw state were analyzed, and five reanalysis datasets of soil moisture and temperature were also evaluated by these in situ datasets. The paper is generally well written and organized. Prior to accepting this paper, the authors may want to address the following issue.**

Thanks very much for your recognition of our work and for your constructive comments. We have carefully considered all the comments into the revision. Please find below the response to each comment for detailed information. We hope that the revised manuscript has appropriately addressed the reviewer's concerns and can be considered for publication in Earth System Science Data.

**General comments:**

1. **It is good to see the authors conducted a detailed evaluation of five reanalysis datasets, i.e., ERA5, GLDAS CLSM, GLDAS Noah, GLDAS VIC, and MERRA2. But in the current version, the authors only present the results which were not interpreted further. For example, why the ERA5 outperforms the other products in simulating the surface soil moisture? Why the model simulations generally underestimate soil moisture at different depths in Maqu network while generally overestimate soil moisture at Shiquanhe network? Why all the model simulated soil temperature at all depths shown a noticeable underestimation in both Maqu and Shiquanhe networks? Some explanations (or discussion) of these results will be helpful to enhance the robustness of the paper.**

Thanks for the comments and suggestions. We have provided additional explanations to the evaluation results of five reanalysis datasets in Section 4.3 of the revised version as shown below.

On Page 15 Line 451-458:

"In general, the modelling uncertainties may be caused by many factors, such as model structure, model parameterization and parameters, and meteorological forcing data. The underestimation of surface SM noted for the Noah, CLSM and MERRA2 can be related to fact that the impact of organic matter on soil hydraulic parameters is ignored (Yi et al., 2011; Chen et al., 2013; Zheng et al., 2015a). The better performance of ERA5 can be associated with the better estimation of precipitation and assimilation of ASCAT SM product (Shi et al., 2021; Hersbach et al., 2020). In the cold season, the Noah SM product generally captures well the $SM_{ups}$ variations at surface layer (i.e., 5 cm) but overestimates the $SM_{ups}$ at deeper layers (e.g., 20 and 40 cm), and overestimations"

are also found for the MERRA2 products at all the depth. The overestimation can be related to the inappropriate parameterization of soil freezing characteristics as shown in Fig. 7 (Zheng et al., 2017)."

On Page 15-16 Line 467-471:

"Similar findings have recently been reported by Ma et al. (2021). The underestimation can be due to the i) underestimation of downward shortwave or longwave radiation (Chen et al., 2011; Wang et al., 2016), ii) inappropriate parameterization of diurnally varying roughness length for heat transfer (Chen et al., 2011; Zheng et al., 2015b; Reichle et al., 2017 ), and iii) overlook of the impact of organic matter on soil thermal parameters (Zheng et al., 2015b)."

On Page 16 Line 487-488:

"This can be related to the underestimation of ST noted for all the model-based products."

On Page 16-17 Line 500-507:

"As in the Maqu network, the better performance of ERA5 can be associated with the better estimation of precipitation and assimilation of ASCAT SM product (Hersbach et al., 2020; Shi et al., 2021). The overestimation noted to other products can be associated with the overestimations of precipitation (Yang et al., 2020) and uncertainty of soil texture and thus overestimation of soil porosity (Su et al., 2013; Shangguan et al., 2013; Bi et al., 2016). Both the Noah and MERRA2 products also overestimate the $SM_{ups}$ of 5 cm in the cold season, which is related to the inappropriate parameterization of soil freezing characteristics as shown in Fig. 10. For the 20 and 40 cm deeper depths, all the products systematically overestimate the $SM_{ups}$ due to uncertainty of soil texture, among which the ERA5 product shows the lowest bias while the VIC product presents the largest bias."

On Page 17 Line 518-519:

"The reason for the underestimation can be the same as the Maqu network."

On Page 17 Line 531-533:

"Similar as the Maqu network, all the model-based products tend to produce earlier onset of freezing and later onset of thawing at every depth due to the underestimation of ST, leading to underestimation of FSD and overestimation of TED and thus longer F/T duration in comparison to the upscaled dataset."

2. **To match the depths of in situ SMST measurements, the model-based SMST data were resampled across the vertical soil profile using the linear interpolation method. Did the authors find the linear interpolation is the best choice after testing or just follows the same procedure in previous studies? Can the uncertainty of this linear interpolation be quantified?**

Thanks for the comments. Linear interpolation and depth-weighted methods were widely used to resample the SMST data across the vertical soil profile in the previous studies. We tested both methods and found the results were similar (see Figs. R1-R2 shown below). In order to make full use of the valid in-situ measurements, we adopted linear interpolation method in this study. The corresponding text have also been added in revised version on Page 9 Line 263-266:

"To match the depths of in-situ SMST measurements, we compared the linear interpolation method and the depth-weighted interpolation method that are widely used to resample the SMST data across the vertical soil profile in the previous studies (Gao et al., 2017), and the results were found to be comparable to each other (Fig not shown). To make full use of the valid in-situ measurements, the linear interpolation method was thus adopted in this study."

Take the GLDAS-2.1 Noah SM product as the example, the calculating process and results of the two methods are shown as below.

- Depth-weighted method

The SMST derived from the GLDAS-2.1 Noah product for the layer 1 (0-10 cm), layer 2 (0-40 cm), and layer 3 (0-80 cm) are calculated as follows:

$$X_{0-10,Noah} = X_{0-10,Noah}$$

$$X_{0-40,Noah} \approx X_{0-10,Noah} * \frac{10}{40} + X_{10-40,Noah} * \frac{30}{40}$$

$$X_{0-80,Noah} \approx X_{0-10,Noah} * \frac{10}{80} + X_{10-40,Noah} * \frac{30}{80} + X_{40-100,Noah} * \frac{40}{80}$$

- Linear interpolation method

The SMST derived from the GLDAS-2.1 Noah product for the depths of 5, 20, and 40 cm are calculated as:

$$X_{5,Noah} \approx X_{0-10,Noah}$$

$$X_{20,Noah} \approx X_{0-10,Noah} + (X_{10-40,Noah} - X_{0-10,Noah}) * (20 - 5)/(25 - 5)$$

$$X_{40,Noah} \approx X_{10-40,Noah} + (X_{40-100,Noah} - X_{10-40,Noah}) * (40 - 25)/(70 - 25)$$

[Figure]

**Figure R1: Time series of daily SM and ST derived from GLDAS-2.1 Noah product at soil depths of 5, 20, and 40 cm and in soil layers of 0-10, 0-40, and 0-80 cm from January 2010 to December 2019 for the Maqu network.**

[Figure]

**Figure R2. Same as Figure R1 but for the Shiquanhe network from January 2011 to December 2018.**

**Specific comments:**

**1. Line 31: it is better to clarify the names of the reanalysis datasets.**

Thanks for the suggestion. The names of the reanalysis datasets have been added in revised version on Page 1 Line 31:

"namely ERA5, MERRA2, GLDAS-2.1 CLSM, Noah, and VIC"

**2. Line 63-64: which climate data and land cover data did you use here?**

Thanks for the comment. The climate classification can be found from Beck et al. (2018), and the land cover data was collected through the field work (Su et al., 2011; Zhang et al., 2021), the references have been added in revised version on Page 3 Line 64-66:

"which are respectively located in the cold humid area with cold dry winter and rainy summer covered by grassland, the cold semiarid area dominated by tundra, and the cold arid area dominated by desert (Su et al., 2011; Beck et al., 2018; Zhang et al., 2021)."

**3. Section 2.2: please give the data portal (where you downloaded the data) here.**

Thanks for the suggestion. The meteorological data portal has been added in revised version on Page 6 Line 167-169:

"Precipitation and air temperature used in this study for the Maqu and Shiquanhe networks are obtained from the meteorological dataset provided by the China Meteorological Administration (http://data.cma.cn/en/?r=data/detail&dataCode=A.0012.0001, last access: 9th September 2022)."

**4. Equation (1): did you conduct regional average vs. regional average validation (that means some averaged model grids did not include in situ data but their values still involved in the spatial averaging)?**

Thanks for the comment. Indeed, we conduct regional average vs. regional average validation. Equation (1) is used to produce the regional-scale SMST based on in-situ measurements, and all the model grids falling into the network area are averaged to obtain the corresponding values from the model-based products. Relevant description is provided in revised version on Page 8 Line 259-262:

"All the model grids falling into the scope of in-situ network are extracted from each product. Afterwards, the native grids of each product are downscaled to 0.25°*0.25° sub-grid cells using a bilinear interpolation. Subsequently, the SMST data in all the sub-grid cells falling into the scope of in-situ network are averaged to match the upscaled in-situ SMST data that represent the regional-scale mean values of in-situ network (see Fig. B1)."

[Figure]

**Figure B1: Grids of the model-based products falling into the (a) Maqu and (b) Shiquanhe network areas (denoted by the colourful dashed rectangles).**

5. **Line 247: did you already consider the difference of coordinated Universal Time (UTC, and note models provide data at UTC) and local solar time (LST)? BTW, did the Tibet-OBS provides data in UTC or LST?**

Thanks for the comment. The Tibet-Obs provides data at Beijing time, we have already considered the difference between UTC and Beijing time (= UTC + 8).

6. **Line 277-279: did you average the monthly mean values for warm season and cold season per year (i.e., for every grid, one value per year for each season)? If not, how did you distinguish the inter-annual and inter-month influences? Moreover, what do you mean by "and all missing data points are assigned an equal value smaller than existed valid data points", why did you do such step?**

Thanks for the comments. Indeed, we use the monthly mean values for every year to calculate the trend, and the warm season spans from May to October while the cold season includes the rest of the months. Please find the details in the Appendix C.

"All missing data points are assigned an equal value smaller than existed valid data points" is one step of the Mann-Kendall trend test. We referred to the book named "Statistical Methods for Environmental Pollution Monitoring" by Gilbert Richland. (1987) Section 16.4 on Page 221:

"This procedure is particularly useful since missing values are allowed and the data need not conform to any particular distribution. Also, data reported as trace or less than the detection limit can be used (if it is acceptable in the context of the population being sampled) by assigning them a common value that is smaller than the smallest measured value in the data set."

The reference has been added in revised version on Page 10 Line 284:

"The Mann Kendall trend test reported by Richland (1987) is used in this study to determine whether a trend is presented within the long-term SMST time series derived either from the upscaled in-situ measurements or from the model-based products."

7. **Line 294: should be "rainfall" rather "snowfall"?**

Thanks for the comment. The "snowfall" is changed to "precipitation" in revised version on Page 11 Line 336.

**8. Line 300: change "diminish" to "diminishes".**

Thanks for the comment. The "diminish" is changed to "diminishes" in revised version on Page 12 Line 343.

**9. Line 319-339, and 365-373, Figs. 4 and 7: it is nice to see the authors investigated the trend of SM and ST in the TP as well as that of some meteorological data. Are there any conclusions about the influencing factors of SM and ST changes in the TP? Are there any similarities and differences in your results (e.g., trend analysis) compared to some similar studies conducted in TP, e.g., Shi et al. (2021, doi: 10.1175/JHM-D-21-0077.1)? What are the possible reasons for these differences? Some discussion of this issue will be helpful for the readability of these parts.**

Thanks for the comments and suggestions. We have provided additional discussion to the trend analysis results compared to the previous study in revised version as shown below.

On Page 12 Line 371-373:

"The full year trend analysis results are consist with the results reported by Shi et al. (2021) using the ESA CCI SM product, since the precipitation is the dominant drive of SM variation which shows significant negative trend in the humid area on the TP."

On Page 14 Line 420-423:

"The result is slightly different from Shi et al. (2021) that might be attributed to the different time span. Nevertheless, it is in agreement with the conclusion of spatial-temporal trend changes of surface SM generally decreasing from southeast to northwest over the TP comparing to the trend analysis result of Maqu network area."

**10. Line 326 and elsewhere: add unit for Sen's slope values, if applicable.**

Thanks for the comment. The unit for Sen's slope values has been added in revised version.

On Page 12 Line 370:

"the Sen's slopes of -0.004 ($m^3 m^{-3}$/yr) and -0.002 ($m^3 m^{-3}$/yr), respectively"

On Page 13 Line 375:

"with a Sen's slope of -0.08 (℃/yr)"

On Page 14 Line 420:

"with a Sen's slope of 0.001 ($m^3 m^{-3}$/yr)"

**11. Line 417-419: was only liquid soil water content in frozen soil simulated by the GLDAS Noah and MERRA2? Any references to illustrate this issue?**

Thanks for the comment. The references has been added in revised version on Page 15 Line 439-440:

"and the MERRA2 and Noah SM products can provide liquid soil water content (Gelaro et al., 2017; Zheng et al., 2017). "

The information of MERRA2 SM data can also refer to https://gmao.gsfc.nasa.gov/reanalysis/MERRA-2/FAQ/#
"FAQ1: WHAT ARE THE MERRA-2 SOIL MOISTURE VARIABLES AND THEIR UNITS?"

**12. Table 4 and elsewhere (e.g., figures, and texts): add unit for Bias, RMSD, and ubRMSD.**

Thanks for the comment. The units for Bias, RMSD, and ubRMSD have been added in Tables 5 and 6 in revised version.

**13. Fig. 3(a): any possible reasons that the SM at 40 cm is the lowest (lower than that at 80 cm)?**

Thanks for the comment. A possible reason has been given in revised version on Page 12 Line 338-340:

"The soil layers below 20 cm are dryer than the upper layers in the warm season, whereas the soil at depth of 80 cm is wetter than 40 cm that might be attributed to absence of evapotranspiration and existence of shallow groundwater (Li et al., 2021 )."

**14. Figs. 5 and 8: the abscissa ST is negative, and the reader may be confused that how did you obtain the thawing results at such situation. Try to make them clearer if possible.**

Thanks for the comment. We have added an explanation of freezing and thawing period in revised version on Page 13 Line 379-381 to make it clearer:

"The freezing period defined in this study spans from the first date of ST falling below zero to the date of lowest ST occur, whereby the SM value is generally decreasing in this period. Later on the thawing period starts and ends when the ST rise above zero, whereby the SM value is increasing during this period."

We would like to thank the reviewer for carefully reading our manuscript and providing detailed and constructive comments. This has helped us to improve our manuscript significantly. In the text below we provide our response to each comment point by point.

Reviewer's comments are in **bold**.

Author's responses are in regular.

Author's additions/modifications in the text are in blue.

**1.    Page 2 Line 58: The authors cited this reference for at least 10 times. How many similarities are there between the cited paper and this paper?**

Thanks for the comment. The similarities between the cited paper and this paper are the adopted methods including spatial upscaling and trend analysis. In addition, the datasets presented in both papers are produced based on the Tibet-Obs measurements. The novelty of this paper in comparison to the cited paper is provided on Page 3 Line 81-85:

"In this paper, we present a long-term (~10 years) SMST profile dataset collected from the Tibet-Obs, which expands the surface SM dataset introduced by Zhang et al. (2021) to include both SM and ST measurements collected at multiple depths. As such, analysis of freezing and thawing characteristics become possible. The analysis of seasonal dynamics and trend changes as well as validation of model-based products are also extended to multiple depths for an approximately 10-year period. In addition, more model-based products are evaluated in this paper."

**2.    Page 2 Line 64: What is short grasses?**

Thanks for the comment. The "short grasses" has been changed to "grassland" in revised version that is well known by readers.

On Page 1 Line 18:

"located in the cold humid area covered by grassland"

On Page 2 Line 65:

"in the cold humid area with cold dry winter and rainy summer covered by grassland"

On Page 4 Line 117:

"with a land cover dominated by grassland"

On Page 18 Line 548:

"in the cold humid area covered by grassland"

**3.    Page 4 Line 112: There should have a chart to clearly summarize the basic properties of each networks.**

Thanks for the comment. A new table, Table 2 in the revised version is added to provide the basic information of the Tibet-Obs networks.

Table 2. Information of the Tibet-Obs networks

| Networks | Climate zone | Land cover | Altitude (m) | Annual Precipitation (mm) | Monitoring sites |
|----------|--------------|------------|--------------|---------------------------|------------------|
| Maqu | Cold humid | Grassland | 3400-3800 | 600 | 26 |

| Shiquanhe Ali | Cold arid | Desert | 4200-4700 | 100 | 20 |
| Naqu | Cold semiarid | Tundra | Around 4500 | 400 | 4 11 |

**4.      Page 5 Line 131: What's the meaning of a.s.l. ?**

Thanks for the comment, it was mentioned on Page 4 Line 117-118: "above sea level (a.s.l)".

**5.      There are numerous of model-based SMST products. Why do you select these ones?**

Thanks for the comment. The reason to select the five model-based products is provided in revised version on Page 6 Line 178-179:

"The reason to select these products is due to the fact that they are more widely adopted and extensively assessed."

**6.      There are many upscaling methods. Why do you use the classic arithmetic averaging approach other than elaborate methods (such as DISPATCH, machine learning, and assimilation). Please explain it.**

Thanks for the comment. The reason to use the classic arithmetic averaging approach is provided in revised version on Page 8 Line 219-223:

"Zhang et al. (2021) demonstrated the better performance of the arithmetic averaging approach in upscaling the surface SM of the Tibet-Obs network in comparison to the voronoi diagrams, time stability, and apparent thermal inertia methods that are widely adopted in existing literatures (Qin et al., 2015; Colliander et al., 2017). Therefore, the arithmetic averaging approach is also adopted in this study to obtain the regional-scale SMST profile data for Maqu and Shiquanhe."

**7.      What's the difference between the soil moisture derived from this paper and Zhang et al. (2021)?**

Thanks for the comment. In Zhang et al. (2021), only the surface SM dataset is presented. In this paper, both profile SM and ST data are presented. Relevant description is provided on Page 3 Line 81-83:

"In this paper, we present a long-term (~10 years) SMST profile dataset collected from the Tibet-Obs, which expands the surface SM dataset introduced by Zhang et al. (2021) to include both SM and ST measurements collected at multiple depths. As such, analysis of freezing and thawing characteristics become possible."

**8.      Page 8 Line 245: How to convert the unit?**

Thanks for the comment. The equation of unit convert has been added in revised version on Page 9 Line 249-253:

"The units of SM data from the GLDAS-2.1 CLSM, Noah, and VIC products is converted from "kg m$^{-2}$" to "m$^3$ m$^{-3}$" following Eq. (2), and the units for the ERA5 and MERRA2 SM data is already with "m$^3$ m$^{-3}$".

$$SM = SWC/(L * \rho_{H_2O}) \tag{2}$$

where $SWC$ represents the soil water content (kg m$^{-2}$), $L$ (m) represents the layer thickness, $\rho_{H_2O}$ represents the soil water density (kg m$^{-3}$)."

**9.      The reliability of upscaled datasets should be verified at first before trend analysis.**

Thanks for the suggestion. We have adjusted the order of Section 4.1 and Section 4.2 in revised version, the corresponding text has been revised on Page 10 Line 293-297:

"Section 4.1 gives the uncertainty analysis results for the upscaled SMST profile data. Section 4.2 presents the upscaled SMST profile data for the Maqu and Shiquanhe networks spanning the 10-year period from 2010 to 2019 (see Section 3.1), as well as the analysis results for the SMST seasonal dynamics, trend test, detection of F/T state and soil freezing characteristics at different depths. Application of the upscaled data to evaluate the performance of model-based products is presented in Section 4.3 to demonstrate its suitability for the evaluation of readily available SMST profile products."

**10.     To prove the uniqueness and superiority of the upscaled dataset for validation, there should have a comparison between the accuracy degrees achieved from the validation against the upscaled and the original dataset, respectively.**

Thanks for your comment. The uniqueness of the SMST dataset from the Tibet-Obs is in terms of its long-term period and multiple soil depths on the Tibetan Plateau. The SMST data used in validating the model-based products is the upscaled dataset, which can generally represent the regional value of monitoring network (i.e. the Maqu and Shiquanhe network areas as shown in Figure B1). Correspondingly, all the model grids falling into the network area are averaged to obtain the regional values as well. This is a common method widely adopted to validate the model-based products to reduce the uncertainty related to the spatial variability of SM. In other words, both of the upscaled in-situ SMST data and averaged model-based SMST data represent the regional condition of network, while the original data is at point-scale for each monitoring site that may exist significant uncertainty for assessment. It is thus pointless to make a comparison between point data and the regional data.

[Figure]

**Figure B1: Grids of the model-based products falling into the (a) Maqu and (b) Shiquanhe network areas (denoted by the colourful dashed rectangles).**

---

## Author Response (AR2)

**Letter to Editor**

Dear Editor,

Thank you very much for reviewing our revised manuscript and providing constructive comments. The manuscript has been revised based on all the comments made by the editor.

Please find below our detailed responses to each comment made by the editor.

We think that the revised manuscript has appropriately addressed all the editor' concerns and we hope that you can consider it for publication in Earth System Science Data.

Sincerely,

Pei Zhang

On behalf of all co-authors

We would like to thank the editor for carefully reading our revised manuscript and providing detailed comments. In the text below we provide our response to each comment point by point.

Editor's comments are in **bold**.

Author's responses are in regular.

Author's additions/modifications in the text are in blue.

**Comments to the author:**

**The reviewers were satisfied with the revision and believed that the concerns and suggestions have been addressed. There are still a few minor errors and issues that need to be fixed before the publication. For example:**

1. **Please carefully check the citations. For example, line 47, (Zheng et al., 2019, 2018), please use time ascending order for the years and put 2018 in the front.**

Thanks for the suggestion. The citations have been checked and corrected using time ascending order in the revised version. For example:

On Page 2 Line 41:

"in the soil-vegetation-atmosphere system (van der Velde et al., 2009; Zheng et al., 2018)."

On Page 2 Line 44:

"and model simulations (Rodell et al., 2004; Entekhabi et al., 2010; Dorigo et al., 2011)."

On Page 2 Line 46-47:

"and model-based products (Zeng et al., 2015; Chen et al., 2017; Colliander et al., 2017;), as well as improving model parametrizations (Zheng et al., 2015a, b, 2017) and remote sensing retrieval algorithms (Zheng et al., 2018, 2019)."

On Page 4 Line 114:

"and additional information about the Tibet-Obs can be found in Su et al. (2011) and Zhang et al. (2021)."

On Page 17 Line 504:

"thus overestimation of soil porosity (Shangguan et al., 2013; Su et al., 2013; Bi et al., 2016)."

**Please check and fix issues with the formations. For example,**

2. **line 201, "The GLDAS-2.1 VIC (Variable Infiltration Capacity) product". Please state the full name first and put the acronyms in the brackets.**

Thanks for the suggestion. All issues of similar kind have been addressed in the revised version. For example:

On Page 6 Line 188:

"The Global Land Data Assimilation System Version 2 Catchment Land Surface Model (GLDAS-2.1 CLSM) product"

On Page 7 Line 201:

"The GLDAS-2.1 Variable Infiltration Capacity (GLDAS-2.1 VIC) product"

On Page 7 Line 208:

"The Modern-Era Retrospective analysis for Research and Applications version 2 (MERRA2)"

**3.  Line 551, "every 15-minute" please remove the minus symbol.**

Thanks for the suggestion. It has been corrected in the revised version:

On Page 4 Line 105:

"record SMST profile dynamics every 15 minutes."

On Page 18 Line 551:

"monitor SMST dynamics at multiple depths (e.g., 5, 10, 20, 40, and 60/80 cm underground) every 15 minutes,"

**There are also issues in the tables and figures. For example:**

**4.  The values in Table 3 are difficult to read because of miss aligning. Please check and improve it if it is possible.**

Thanks for the suggestion. Table 3 has been improved in the revised version to make the information more clear.

Table 3. Information for the selected model-based products.

| Product | Spatial Resolution | Temporal Resolution | Temporal Coverage | SM Stratification (cm) | ST Stratification (cm) |
|---------|--------------------|---------------------|-------------------|------------------------|------------------------|
| ERA5 | 0.25°× 0.25° | Hourly | 1979 ongoing | 0-7, 7-28, 28-100, 100-289 | |
| Noah | 0.25°× 0.25° | 3 Hours | 2000 ongoing | 0-10, 10-40, 40-100 | |
| CLSM | 1°×1° | 3 Hours | 2000 ongoing | 0-2, 0-100 | 0-10, 10-29, 29-68, 68-144 |
| VIC | 1°×1° | 3 Hours | 2000 ongoing | 0-30, 30-130*, 130-150* | |
| MERRA2 | 0.5°×0.625° | Hourly | 1980 ongoing | 0-5*, 0-100* | 0-10*, 10-30*, 30-70*, 70-146* |

* The depth of this layer varies with region, and the value shown here is for our study area.

**5.  Table 5 and 6, please explain the use of symbol "-".**

Thanks for the suggestion. The symbol "-" has been explained below the Table 5 in the revised version:

" '-' represents there is not statistic for the ERA5, CLSM, and VIC products in the cold season."

**6.  Figure 1, please add a description or caption to the main legend to help with interpretation.**

Thanks for the suggestion. Description has been added into the caption of Figure 1 in the revised version:

"Figure 1. (a) Location of the Tibet-Obs network over the TP; Spatial distributions of SMST monitoring sites and weather station within the (b) Maqu, (c) Ali, (d) Shiquanhe, and (e) Naqu networks; and (f) an example of instruments configured for each SMST monitoring site. The elevation of the network is shown at the bottom of each subplot. The triangles with different colours represent the SMST measured at different depths. (Base map is from EROS, Copyright: © EROS)"

**7.  Figure 2, add a comma before the last and in the caption.**

Thanks for the suggestion. The comma has been added in the caption of Figure 2 in the revised version:

"Figure 2. Number of available SMST monitoring sites for different depths at each month for the (a) Maqu, (b) Shiquanhe, (c) Ali, and (d) Naqu networks."

**8. Figure 13, the labels in the sub plots c and f seemed duplicated because the lines were already explained in legend.**

Thanks for the suggestion. The labels in the subplots have been removed from Figures 11 and 13 in the revised version:

[Figure]

**Figure 11. Time series of daily average SM (a-c) and monthly mean ST (d-f) at soil depths of 5 (a, d), 20 (b, e), and 40 cm (c, f) derived from the upscaled SMST dataset and five model-based products from January 2010 to December 2018 for the Maqu network.**

[Figure]

**Figure 13. Same as Figure 11 but for the Shiquanhe network from January 2011 to December 2018.**

**9. Captions of Figure A2 and A3. There is no Table A1 provided in the manuscript. Did you mean Figure A1. Please check.**

Thanks for the comment. The "Table A1" has been changed to "Figure A1" in the caption of Figures A2 and A3:

"Figure A2. Same as Figure A1 but for the Ngari network"

"Figure A3. Same as Figure A1 but for the Naqu network"